# Delaunay Component Analysis for Evaluation of Data Representations

**Petra Poklukar,**[*] **Vladislav Polianskii, Anastasia Varava, Florian T. Pokorny & Danica Kragic**
KTH Royal Institute of Technology,
Stockholm, Sweden

## Abstract

Advanced representation learning techniques require reliable and general evaluation methods. Recently, several algorithms based on the common idea of geometric and topological analysis of a manifold approximated from the learned data representations have been proposed. In this work, we introduce Delaunay Component Analysis (DCA) – an evaluation algorithm which approximates the data manifold using a more suitable neighbourhood graph called Delaunay graph. This provides a reliable manifold estimation even for challenging geometric arrangements of representations such as clusters with varying shape and density as well as outliers, which is where existing methods often fail. Furthermore, we exploit the nature of Delaunay graphs and introduce a framework for assessing the quality of individual novel data representations. We experimentally validate the proposed DCA method on representations obtained from neural networks trained with contrastive objective, supervised and generative models, and demonstrate various use cases of our extended single point evaluation framework.

## 1 Introduction

Quality evaluation of learned data representations is gaining attention in the machine learning community due to the booming development of representation learning techniques. One common approach is to assess representation quality based on their performance on a pre-designed downstream task (Bevilacqua et al., 2021; Li et al., 2020). Typically, a classification problem is used to evaluate either the ability of a model to recover labels of raw inputs, or the transferability of their representations to other domains, as done in state-of-the-art unsupervised representation learning methods (Chen et al., 2020b; Ermolov et al., 2021). However, in many scenarios such straightforward downstream classification task cannot be defined, for instance, because it does not represent the nature of the application or due to the scarcity of labeled data as often occurs in robotics (Chamzas et al., 2021; Lippi et al., 2020). In these scenarios, representations are commonly evaluated on hand-crafted downstream tasks, e.g., specific robotics tasks. However, these are time consuming to design, and potentially also bias evaluation procedures and consequentially hinder generalization of representations across different tasks.

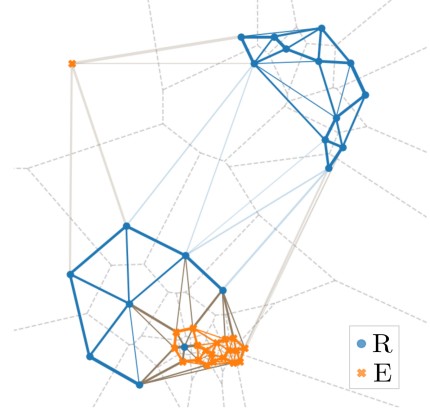

Figure 1: Example of an approximated Delaunay graph $\mathcal{G}_D = \mathcal{G}_D(R \cup E)$ (solid edges) obtained from the Voronoi cells (dashed gray lines) of the considered $R$ and $E$ points as well as the distilled Delaunay graph $\mathcal{G}_{DD}$ containing three connected components (solid dark colored and gray edges) used in the final evaluation of $R$ and $E$. See Section 3 for furher details.

Recently, more general evaluation methods such as Geometry Score (GS) (Khrulkov & Oseledets, 2018), Improved Precision and Recall (IPR) (Kynkäänniemi et al., 2019) and Geometric Component Analysis (GeomCA) (Poklukar et al., 2021) have been proposed.

---

[*]Correspondence to Petra Poklukar, poklukar@kth.se.

These methods analyze *global geometric and topological properties* of representation spaces instead of relying on specific pre-designed downstream tasks. These works assume that a set of evaluation representations $E$ is of high quality if it closely mimics the structure of the true data manifold captured by a reference set of representations $R$. This reasoning implies that $R$ and $E$ must have similar geometric and topological properties, such as connected components, their number and size, which are extracted from various approximations of the data manifolds corresponding to $R$ and $E$. For example, GS and GeomCA estimate the manifolds using simplicial complexes and proximity graphs, respectively, while IPR leverages a $k$-nearest neighbour ($k$NN) based approximation. However, as we discuss in Section 2, neither of these approaches provides a reliable manifold estimate in complex arrangements of $R$ and $E$, for instance, when points form clusters of varying shape and density as well as in the presence of outliers. Moreover, the informativeness of the scores introduced by these methods is limited.

In this work, we address the impediments of evaluation of learned representations arising from poor manifold approximations by relying on a more natural estimate using Delaunay graphs. As seen in Figure 1, edges (solid lines) in a Delaunay graph connect *spacial* neighbours and thus vary in length. In this way, they naturally capture local changes in the density of the representation space and thus more reliably detect outliers, all without depending on hyperparameters. We propose an evaluation framework called Delaunay Component Analysis (DCA) which builds the Delaunay graph on the union $R \cup E$, extracts its connected components, and applies existing geometric evaluation scores to analyze them. We experimentally validate DCA on a variety of setups by evaluating contrastive representations (Section 4.1), generative models (Section 4.2) and supervised models (Section 4.3).

Furthermore, we exploit the natural neighbourhood structure of Delaunay graphs to evaluate a single *query* representation. This is crucial in applications with continuous stream of data, for example, in interactions of an intelligent agent with the environment or in the assessment of the visual quality of individual images generated by a generative model as also explored by Kynkäänniemi et al. (2019). In these cases, existing representation spaces need to be updated either by embedding novel query points or distinguishing them from previously seen ones. In Delaunay graphs, this translates to analyzing newly added edges to the given query point. We demonstrate various possible analyses in Section 4 using aforementioned experimental setups.

## 2   STATE-OF-THE-ART METHODS AND THEIR LIMITATIONS

We review the state-of-the-art methods for evaluation of learned data representations, namely GS (Khrulkov & Oseledets, 2018), IPR (Kynkäänniemi et al., 2019) and GeomCA (Poklukar et al., 2021), which compare topological and geometrical properties of an evaluation set $E$ with a reference set $R$ representing the true underlying data manifold. We discuss differences in their manifold approximations, visualized in Figure 2, as well as informativeness of their scores.

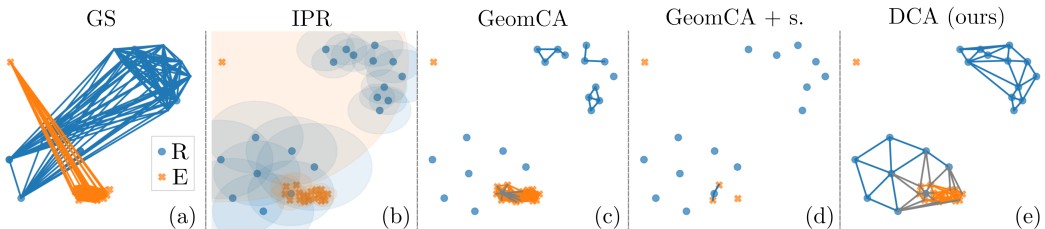

Figure 2: Visualization of manifold approximations obtained by GS (a), IPR (b), GeomCA without sparsification (c), GeomCA with sparsification (d) and our proposed DCA (e).

The pioneering method in this area, GS, constructs witness simplicial complexes on randomly sampled subsets of $R$ and $E$ (panel (a)), and compares their connected components using tools of computational topology (Zomorodian & Carlsson, 2004). The result is an average over several iterations summarized either in the form of histograms, which are cumbersome for quantitative comparison, or as a single un-normalized score, which is in many cases not sufficiently informative. Moreover, GS

depends on four hyperparameters, which can be difficult to understand and tuned by practitioners unfamiliar with computational topology.

In contrast, IPR obtains manifold approximations by enlarging each point in $R$ (or $E$) with a hypersphere of radius equal to the distance to its $k$NN in that set (panel (b)). It defines two scores: *precision* $\mathcal{P}^I$ which counts the number of $E$ points contained on the approximated $R$ manifold, and vice versa for *recall* $\mathcal{R}^I$. While the method depends only on one hyperparameter $k$, it is highly affected by outliers which induce overly large spheres and dense areas of the space which yield too conservative coverage as visualized in panel (b). Moreover, it is the only method expecting $R$ and $E$ to be of equal size, thus, often requiring subsampling of one of them. Both GS and IPR have been primarily developed to assess generative adversarial networks (GANs) (Goodfellow et al., 2014).

The most recent and generally applicable method, GeomCA, extends GS and IPR in two ways: *i)* it provides functionality to analyze individual connected components of the approximated manifold, hence enabling one to *examine local areas* of representation spaces where inconsistencies arise, and *ii)* characterizes the captured geometry in four global scores: *precision* $\mathcal{P}^G$ and *recall* $\mathcal{R}^G$, which are similar to IPR, as well as *network consistency* $c^G$ *and network quality* $q^G$, which are also the basis of the GeomCA local evaluation scores (see Section 3 for recap). GeomCA estimates the manifolds using $\varepsilon$-proximity graphs where the hyperparameter $\varepsilon$ defines the maximum distance between two points connected by an edge and is estimated from distances among points in $R$. However, in practice, representations often form clusters of different shape and density which makes it impossible to select a single value of $\varepsilon$ that adequately captures such variations (see examples in panel (c)).

Moreover, GeomCA reduces the number of points by performing geometric sparsification that extracts a subset of points from each of $R$ and $E$ being pairwise at least distance $\delta$ apart, where $\delta$ is estimated from $\varepsilon$. While the authors argue that this step does not affect their scores as it preserves the topology, we show that it nevertheless can bias the manifold estimation of $R$ and $E$. An example is illustrated in the bottom component of panels (c) and (d), where the sparsification removes a large portion of the dense $E$ subset and artificially increases the precision. We demonstrate the occurrence of this scenario in the evaluation of a GAN model in Section 4.2.

Our DCA framework utilizes Delaunay graphs (panel (e)) to address the discussed limitations in manifold approximations of the existing methods, and employs evaluation scores introduced by Poklukar et al. (2021) to maximize its informativeness. Moreover, it extends the existing methods by additionally providing a general framework for quality evaluation of single query representations.

## 3 METHOD

We propose *Delaunay Component Analysis (DCA)* algorithm for evaluation of data representations consisting of three parts: *i) manifold approximation* which approximates the Delaunay graph on the given sets of representations, *ii) component distillation* which distills the graph into connected components, and lastly *iii) component evaluation* which outputs the evaluation scores summarizing the geometry of the data. We provide an outline of our DCA framework in Algorithm 1 found in Appendix A. Moreover, by exploiting the nature of Delaunay graphs, DCA can be efficiently implemented (Section 3.2) and extended for evaluation of individual representations (Section 3.1).

**Phase 1: Manifold approximation** As mentioned in Section 1, the unique property of Delaunay graphs is the definition of neighbouring points that are connected by an edge. For example, in an $\varepsilon$-graph, two points are adjacent if they are at most $\varepsilon$ distance apart. In a $k$NN graph, a point is connected to all points that are closer than the $k$-th smallest distance of that point to any other. In contrast, in a Delaunay graph (Figure 1), a point is adjacent to any point that is its spatial neighbour, regardless of the actual distance between them or the number of its other spatial neighbours. We refer to such spatial neighbours as *natural* neighbours of a point and rigorously define them through Voronoi cells (depicted as dashed lines in Figure 1) in the following.

**Definition 3.1** *Given a set* $W \subset \mathbb{R}^N$ *we define the Voronoi cell* $Cell(z)$ *associated to a point* $z \in W$ *as the set of points in* $\mathbb{R}^N$ *for which* $z$ *is the closest among* $W$: $\mathrm{Cell}(z) = \left\{ x \in \mathbb{R}^N \mid \|x - z\| \le \|x - z_i\| \forall z_i \in W \right\}$. *The Delaunay graph* $\mathcal{G}_D(W) = (\mathcal{V}, \mathcal{E})$ *built on the set* $\mathcal{V} = W$ *is then defined by connecting points whose Voronoi cells intersect, i.e.,* $\mathcal{E} = \left\{ (z_i, z_j) \in W \times W \mid \mathrm{Cell}(z_i) \cap \mathrm{Cell}(z_j) \neq \emptyset, z_i \neq z_j \right\}$.

We consider a reference set $R = \{z_i\}_{i=1}^{n_R} \subset \mathbb{R}^N$ and evaluation set $E = \{z_i\}_{i=1}^{n_E} \subset \mathbb{R}^N$ of data representations with $R \neq E$, and approximate the Delaunay graph $\mathcal{G}_D = \mathcal{G}_D(R \cup E)$. By Definition 3.1, edges in $\mathcal{G}_D$ correspond to points on the boundary of Voronoi cells which are obtained using Monte Carlo based sampling algorithm presented by Polianskii & Pokorny (2019) (see Appendix A for further details). The process, visualized in Figure 3, is based on sampling rays originating from each $z \in R \cup E$ and finding their intersection with the boundary of its Voronoi cell $\text{Cell}(z)$. This allows to reconstruct $\mathcal{G}_D$ via subgraph approximation, and can be additionally exploited for memory optimizations which we present in Section 3.2. Due to the sampling, the number of found edges directly depends on the number $T$ of rays sampled from each $z$. However, as we show in ablation studies in Appendix B.1 and B.2, our evaluation framework is stable with respect to the variations in $T$. Next, we discuss the distillation of $\mathcal{G}_D$ into connected components.

**Phase 2: Component distillation** Since edges in $\mathcal{G}_D$ are obtained among natural neighbours, $\mathcal{G}_D$ consists of a single connected component uniting regions of points formed in different densities or shape. These can be distinguished by removing large edges (depicted with transparent edges in Figure 1), or equivalently, by finding clusters of points having similar natural neighbours (depicted with opaque edges in Figure 1).

We distill $\mathcal{G}_D$ into connected components by adapting the state-of-the-art hierarchical clustering algorithm HDBSCAN (McInnes et al., 2017) summarized in Appendix A. We apply the part of HDBSCAN that extracts connected components $\{\mathcal{G}_i\}$ from the minimum spanning tree[1] $\text{MST}(\mathcal{G}_D)$. We emphasise that applying HDBSCAN directly on $\text{MST}(\mathcal{G}_D)$ efficiently bypasses the calculation of a complete pairwise distance matrix[2] of size $n_R + n_E$ which becomes a computational burden for large $R$ and $E$ sets. Such calculation is an integral part of HDBSCAN performed to reduce the sensitivity of the method to noise or outliers which can be omitted in case of Delaunay graphs due their natural neighbourhood structure (see Appendix A for further details). In this way, our modification of HDBSCAN inherits only one of its original hyperparameters, the minimum cluster size $mcs$ parameter, determining the minimum number of points needed for a set of points to form a cluster. This parameter is intuitive to tune and can be flexibly adjusted depending on the nature of the application. In our ablation studies reported in

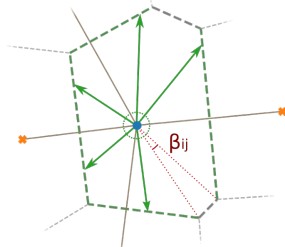

Figure 3: Delaunay graph approximation. In this example, five rays (green arrows) detect four Delaunay edges (solid gray). The angular size $\beta_{ij}$ of the bottom right boundary is marked in red.

Appendix B.1 and B.2, we show that DCA is stable with respect to variations in $mcs$. At the end of this phase, we obtain the *distilled Delaunay graph* $\mathcal{G}_{DD} = \bigsqcup_i \mathcal{G}_i$ of $\mathcal{G}_D$ which we denote simply by $\mathcal{G}$ when no confusion arises. Lastly, we analyze the components of $\mathcal{G}$ as summarized below.

**Phase 3: Component evaluation** We analyze the connected components $\mathcal{G}_i$ of $\mathcal{G}$ using local and global evaluation scores introduced by Poklukar et al. (2021) which we recap below. Following their notation, we denote by $|\mathcal{G}|_\mathcal{V}$ and $|\mathcal{G}|_\mathcal{E}$ the cardinalities of the vertex and edge sets of a graph $\mathcal{G} = (\mathcal{V}, \mathcal{E})$, respectively, and by $\mathcal{G}^Q = (\mathcal{V}|_Q, \mathcal{E}|_{Q \times Q}) \subset \mathcal{G}$ its restriction to a set $Q \subset \mathcal{V}$.

We start by introducing the two local scores: *component consistency and quality*. Intuitively, a component $\mathcal{G}_i$ attains high consistency if it is equally represented by points from $R$ and $E$, i.e., if $|\mathcal{G}_i^R|_\mathcal{V} \approx |\mathcal{G}_i^E|_\mathcal{V}$, where $\mathcal{G}_i^R, \mathcal{G}_i^E$ denote the restrictions of $\mathcal{G}_i$ to $R$ and $E$. Similarly, $\mathcal{G}_i$ obtains a high quality score if points from $R$ and $E$ are also well mixed which is measured in terms of edges connecting $R$ and $E$, i.e., the points are geometrically well aligned if the number of *homogeneous* edges among points in each of the sets, $|\mathcal{G}_i^R|_\mathcal{E}$ and $|\mathcal{G}_i^E|_\mathcal{E}$, is small compared to the number of *heterogeneous* edges connecting representations from $R$ and $E$. This is rigorously defined as follows:

**Definition 3.2 (Local Evaluation Scores, Poklukar et al. (2021))** *Consistency $c$ and quality $q$ of a component $\mathcal{G}_i \subset \mathcal{G}$ are defined as the ratios*

$$c(\mathcal{G}_i) = 1 - \frac{|\,|\mathcal{G}_i^R|_\mathcal{V} - |\mathcal{G}_i^E|_\mathcal{V}\,|}{|\mathcal{G}_i|_\mathcal{V}} \quad and \quad q(\mathcal{G}_i) = \begin{cases} 1 - \frac{(|\mathcal{G}_i^R|_\mathcal{E} + |\mathcal{G}_i^E|_\mathcal{E})}{|\mathcal{G}_i|_\mathcal{E}} & if\ |\mathcal{G}_i|_\mathcal{E} \geq 1, \\ 0 & otherwise, \end{cases} \quad (1)$$

---

[1]Minimum spanning tree of a graph is a tree minimizing the total edge length and connecting all vertices.
[2]The matrix represents mutual reachability distance calculated among each pair of points with respect to the minimum samples parameter.

*respectively. Moreover, $\mathcal{G}_i$ is called consistent if $c(\mathcal{G}_i) > \eta_c$ and of high-quality if $q(\mathcal{G}_i) > \eta_q$ for given thresholds $\eta_c, \eta_q \in [0, 1) \subset \mathbb{R}$. A consistent component of high-quality as determined by $\eta_c, \eta_q$ is called a fundamental component. The union of fundamental components is denoted by $\mathcal{F} = \mathcal{F}(\mathcal{G}, \eta_c, \eta_q)$ and indexed by the subscript $f$, i.e., we write $\mathcal{G}_f \in \mathcal{F}$.*

The thresholds $\eta_c, \eta_q$ are designed to enable a flexible definition of fundamental components and are assumed to be set depending on the application and available prior knowledge. By examining the proportion of $R$ and $E$ points contained in $\mathcal{F}$, we obtain the first two global scores: *precision* and *recall*, respectively. Two more global scores, *network consistency* and *network quality*, used to measure global imbalances and misalignment between $R$ and $E$, can be simply derived by extending Definition 3.2 to the entire graph $\mathcal{G}$. In summary, we consider the following global evaluation scores:

**Definition 3.3 (Global Evaluation Scores, Poklukar et al. (2021))** *We define network consistency $c(\mathcal{G})$ and network quality $q(\mathcal{G})$ as in Definition 3.2, as well as precision $\mathcal{P}$ and recall $\mathcal{R}$ as*

$$\mathcal{P} = \frac{|\mathcal{F}^E|_\mathcal{V}}{|\mathcal{G}^E|_\mathcal{V}} \quad and \quad \mathcal{R} = \frac{|\mathcal{F}^R|_\mathcal{V}}{|\mathcal{G}^R|_\mathcal{V}}, \tag{2}$$

*respectively, where $\mathcal{F}^R, \mathcal{F}^E$ denote the restrictions of $\mathcal{F} = \mathcal{F}(\mathcal{G}, \eta_c, \eta_q)$ to the sets $R$ and $E$.*

### 3.1 EXTENSION: QUERY POINT INSERTION AND EVALUATION

A practical evaluation framework should not only provide a functionality to analyse *sets* of representations but also to assess the quality of *individual* points with respect to $R$, which is desirable in any application where data is being collected continuously. In case of Delaunay graphs, evaluating the quality of a query point $q \subset \mathbb{R}^N \backslash R$ is equivalent to studying its insertion into the distilled Delaunay graph $\mathcal{G}_{DD}(R)$. Formally, we consider the *neighbourhood* $N(q)$ of $q$ in the distilled Delaunay graph $\mathcal{G}_{DD}(R \cup \{q\})$ defined as the subgraph of $\mathcal{G}_D(R \cup \{q\})$ induced by all vertices adjacent to $q$ including $q$ and intersected with $\mathcal{G}_{DD}(R)$. To find the edges of $N(q)$, it suffices to perform the sampling process described in Phase 1 only within the Voronoi cell $\text{Cell}(q)$ of $\mathcal{G}_D$. In fact, this process can be efficiently performed for a set $Q$ of query points independently from each other and has equal computational complexity as the construction of $\mathcal{G}_D(R)$ provided that $|R|$ and $|Q|$ are comparable. We refer to this extension as *query (point) Delaunay Component Analysis*, or in short, *q-DCA* algorithm.

The obtained neighbourhood graph $N(q) = (\mathcal{V}^{N(q)}, \mathcal{E}^{N(q)})$ can be analyzed in numerous ways depending on the application. In this work, we experiment with the length and the number of the inserted edges in three ways: *i)* by extracting the point in $R$ closest to $q$, *ii)* by additionally extracting the length of that edge, or *iii)* by examining the number of inserted edges to each fundamental component as well as the length of the shortest one. We provide a summary of the three variants in Algorithm 2 in Appendix A and experimentally validate them in Section 4.

### 3.2 DELAUNAY GRAPH REDUCTION VIA SPHERE COVERAGE

In high dimensions, the Delaunay graph $\mathcal{G}_D$ can have a large number of edges. In this section, we propose an optional procedure for removing the least significant edges and thus reducing the overall memory consumption.

During the construction of $\mathcal{G}_D$, the probability of sampling an edge $(z_i, z_j) \in \mathcal{E}(\mathcal{G}_D)$ directly depends on the angular size of $\text{Cell}(z_i) \cap \text{Cell}(z_j)$ viewed from one of the points[3]. It is visualized in Figure 3 and formally defined as the solid angle $\beta_{ij} \in (0, 1]$ at $z_i$ of a cone with vertex $z_i$ and base $\text{Cell}(z_i) \cap \text{Cell}(z_j)$ normalized by the volume of the unit hypersphere. It is approximated by the ratio of all rays sampled from $z_i$ that provided the corresponding edge $(z_i, z_j)$ using the algorithm of Polianskii & Pokorny (2019). Intuitively, low $\beta_{ij}$ values correspond to edges that are unstable (depicted with thinner lines in Figure 1) because they may disappear with small perturbations of data. We therefore propose to remove the longest edges $(z_i, z_j)$ connecting $z_i$ such that the sum $\sum_j \beta_{ij}$ of solid angles corresponding to the remaining ones is larger than a predetermined parameter $B \in [0, 1)$. We refer to $B$ as *sphere coverage* and demonstrate its usefulness on the large scale experiment in Section 4.2 and perform an ablation study in Appendix B.1.

---

[3]The authors of Rushdi et al. (2017) call this significance.

### 3.3 COMPLEXITY ANALYSIS

The computational complexity of the Delaunay graph approximation performed in Phase 1 grows polynomially with the dimensionality of representations $N$. Given that the number of points $|R| + |E|$ is at least linear corresponding to their dimensionality $N$, i.e. $|R| + |E| = \Omega(N)$, we utilize the probabilistic method for the construction of the graph proposed by Polianskii & Pokorny (2019) that yields the total complexity $\mathcal{O}((|R| + |E|)^2 \cdot (N + T))$. The complexity for $q$-DCA is obtained by substituting $|E|$ with $|Q|$ for a set of query points $Q$. Moreover, the asymptotics of the modified HDBSCAN clustering algorithm utilized in Phase 2 rely on the graph obtained in Phase 1, which again does not directly depend on the dimensionality $N$ and is at most quadratic of the number of points, i.e. $\mathcal{O}((|R| + |E|)^2)$. Real computation times of Phase 1 and Phase 2 are comparable (see Appendix B.1, Table 6), partially due to high GPU parallelization of Phase 1. We refer the reader to (Polianskii & Pokorny, 2019), (Campello et al., 2015) and (McInnes & Healy, 2017) for details.

## 4 EXPERIMENTS

Our implementation of the DCA algorithm is based on the C++/OpenCL implementation of Delaunay graph approximation provided by Polianskii & Pokorny (2019) as well as on Python libraries HDBSCAN (McInnes et al., 2017) and Networkx (Hagberg et al., 2008). The code is available on Github[4]. We considered a similar experimental setup as in (Poklukar et al., 2021) and analyzed *(i)* representation space of a contrastive learning model trained with NT-Xent contrastive loss (Chen et al., 2020a), *(ii)* generation capabilities of a StyleGAN trained on the FFHQ dataset (Karras et al., 2019), and *(iii)* representation space of the widely used VGG16 supervised model (Simonyan & Zisserman, 2015) pretrained on the ImageNet dataset (Deng et al., 2009).

Using each of the above scenarios, we demonstrate the stability and informativeness of DCA and showcase three variants of the q-DCA extension. We compare the obtained results to the discussed evaluation methods: GeomCA used with geometric sparsification (denoted by $\mathcal{P}^{sG}, \mathcal{R}^{sG}, c^{sG}, q^{sG}$) and without it (denoted by $\mathcal{P}^G, \mathcal{R}^G, c^G, q^G$), IPR (denoted by $\mathcal{P}^I, \mathcal{R}^I$) and GS. We refer to the evaluation scores obtained using the Delaunay graph as *DCA scores*. We report the hyperparameters choices of the respective methods in Appendix B.

### 4.1 CONTRASTIVE REPRESENTATIONS

A successfully trained contrastive learner provides a controlled setup with known structure of learned representations, thus enabling us to perform a series of experiments validating the correctness and reliability of our DCA method. In this section, we demonstrate the superior performance of DCA compared to the benchmark methods in various complex geometric arrangements of $R$ and $E$. Additional experiments on this experimental setup can be found in Appendix B.1, where we *i)* show how q-DCA can be applied to investigate the robustness of the model's representation space to novel images, *ii)* demonstrate that DCA successfully recognizes mode collapse and mode discovery situations by repeating the mode truncation experiment performed by Poklukar et al. (2021), and lastly *iii)* perform a thorough ablation study on hyperparameters of DCA.

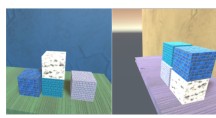

We used a model trained on an image dataset shown in Figure 4 presented by Chamzas et al. (2021). The training $\mathcal{D}_f^R$ and test $\mathcal{D}_f^E$ datasets each consisted of 5000 images containing four boxes arranged in 12 possible configurations, referred to as classes, recorded from the front camera angle (Figure 4 left). We created the sets $R$ and $E$ from 12-dimensional encodings of $\mathcal{D}_f^R$ and $\mathcal{D}_f^E$, respectively, corresponding to the first 7 classes, $c_0, \ldots, c_6$. Due to the contrastive training objective, we expect to observe 7 clusters corresponding to each class and therefore set $\eta_c, \eta_q$, defining fundamental components, to rather high values of $0.75$ and $0.45$, respectively.

Figure 4: Box images recorded from the front and right views, respectively.

**Varying component density** To demonstrate reliability of DCA in various geometric arrangements of points, we repeatedly sampled 10 times 3 classes $c_i$ and discarded a fraction of $p \in \{0.5, 0.75, 0.999\}$ points in $R$ with that label. In this way, $R$ contained smaller and sparser components for $p = 0.5, 0.75$, while $p = 0.999$ mimics a scenario with outliers. Since such pruning

[4]https://github.com/petrapoklukar/DCA

necessarily removed 3 fundamental components due to $\eta_c = 0.75$, we expected to observe the average precision $\mathcal{P} \approx 4/7$ regardless the value of $p$. On the contrary, increasing $p$ corresponds to majority of $R$ being contained in the remaining 4 fundamental components, thus $\mathcal{R}$ should approach 1 (see Appendix B.1 for precise calculation of the optimal values). Since by design, $|R| < |E|$, we performed downsampling of $E$ in case of IPR.

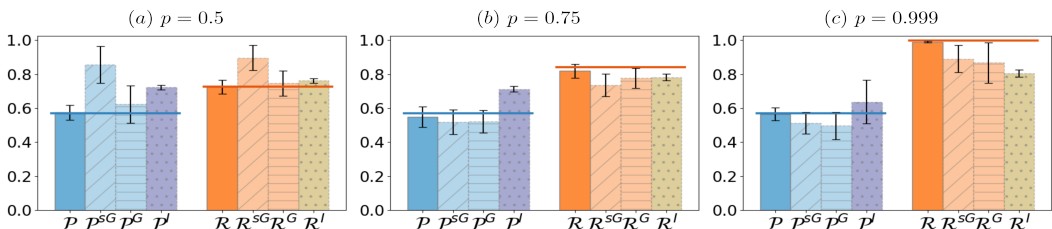

Figure 5: Mean and std of DCA $(\mathcal{P}, \mathcal{R})$, GeomCA with and without sparsification $(\mathcal{P}^{sG}, \mathcal{R}^{sG}, \mathcal{P}^{G}, \mathcal{R}^{G})$, and IPR scores $(\mathcal{P}^{I}, \mathcal{R}^{I})$ obtained when varying the fraction $p$ controlling the amount of discarded $R$ points. Vertical lines denote optimal values for $\mathcal{P}$ (blue) and $\mathcal{R}$ (orange).

Figure 5 shows the mean and standard deviation (std) of the obtained DCA, GeomCA and IPR scores together with the optimal values (solid vertical lines). For DCA, we observe the correct average $\mathcal{P} \approx 0.57$ for all $p$ as well as increasing $\mathcal{R}$ with increasing $p$. Moreover, we observed the scores to have a low std indicating the stability of our method. For $p = 0.5$, the sparsification in GeomCA led to an artificial increase in scores, while $\mathcal{P}^{I}, \mathcal{R}^{I} \approx 0.7$ did not detect the differences in the sets potentially due to the additional downsampling. While the average GeomCA scores obtained without sparsification are more aligned with DCA, they exhibit large std due to the unstable $\varepsilon$ estimation. Similar conclusions hold for $p = 0.75$ and $p = 0.999$, where both GeomCA and sparse GeomCA scores are lower than DCA due to a largely disconnected graph resulting from a small $\varepsilon$. The IPR was again not robust for $p = 0.75$, while in $p = 0.999$, $\mathcal{P}^{I}$ exhibited larger mean and std compared to other scores indicating the erroneous merges originating from large radii of the outliers. Lastly, downsampling of $E$ negatively affected $\mathcal{R}^{I}$ resulting in a low score. In Appendix B.1, we also report the results for GS which we found to be inadequaltely informative.

## 4.2 EVALUATION OF A STYLEGAN

Next, we applied our proposed DCA and q-DCA algorithms to assess the quality of both data distribution learned by a StyleGAN model trained on the FFHQ dataset and individual generated images.

**Generation capacity** To evaluate generation capabilities of StyleGAN, we repeated the truncation experiment performed in (Kynkäänniemi et al., 2019; Poklukar et al., 2021) where during testing latent vectors were sampled from a normal distribution truncated with a parameter $\psi$ affecting the perceptual quality and variation of the generated images. A lower $\psi$ results in improved quality of individual images but at the cost of reducing the overall generation variety. For each truncation $\psi \in \{0.0, 0.1, \ldots, 1.0\}$, we randomly sampled 50000 training images and generated 50000 images, and encoded them into a VGG16 model pretrained on ImageNet. The corresponding 4096-dimensional representations composed the sets $R$ and $E$, respectively. For low values of $\psi$, we expect to obtain high precision values as the high-quality generated images should be embedded close to the (fraction of) training images but low recall values as the former poorly cover the diversity of the training images. Due to the large number of considered representations, we reduced the size of the approximated Delaunay graph $G_D(R \cup E)$ by setting the sphere coverage parameter $B$, introduced in Section 3.2, to 0.7.

Figure 6 shows the obtained DCA scores (solid lines) together with GeomCA, IPR and GS scores (dashed lines). All methods correctly reflect the truncation level $\psi$ apart from GS at $\psi = 0$ and DCA at $\psi = 0.5$, which we discuss in detail below. Firstly, DCA scores exhibit larger absolute values of $\mathcal{P}, \mathcal{R}$ compared to the benchmarks. Even though this is expected because Delaunay edges naturally connect more points, we additionally investigated the average length of homogeneous edges among $R$ and among $E$ points to verify the correctness of the obtained scores. In Table 7, we observe that the average length homogeneous edges among $E$ increases with increasing $\psi$ and resembles the average length homogeneous edges among $R$ for $\psi = 1.0$. This indicates that the StyleGAN model

was successfully trained, which is on par with the high-quality images reported by Karras et al. (2019). However, both GeomCA and IPR methods result in more conservative values which are especially visible for recall. For GeomCA, this is because $\varepsilon$ is estimated from $R$ resulting in fewer and smaller components (also due to the sparsification), which in turn yields low $\mathcal{P}^{sG}, \mathcal{R}^{sG}$. For IPR, it is because the radii of spheres around $E$ points yield a conservative coverage as discussed in Section 2. Therefore, this analysis demonstrates that DCA scores in this case provide more accurate description of the geometry.

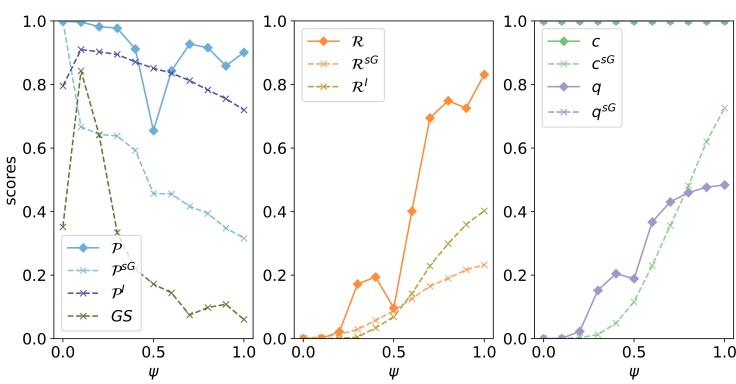

Figure 6: Mode truncation results. Left: DCA $\mathcal{P}$, GeomCA $\mathcal{P}^{sG}$ and IPR $\mathcal{P}^{I}$ precision scores (in blue colors), together with GS (green). Middle: DCA $\mathcal{R}$, GeomCA $\mathcal{R}^{sG}$ and IPR $\mathcal{R}^{I}$ recall scores. Right: DCA $(c, q)$ and GeomCA $(c^{sG}, q^{sG})$ network consistency (light green) and quality scores (purple).

Next, we analyze the inconsistency in $R$ and $E$ detected by DCA at $\psi = 0.5$. It originates from their peculiar geometric position similar to that visualized in Figure 2, where points in $E$ formed a denser cluster that was slightly mixed but mostly concatenated to a more scattered cluster in $R$. As discussed in Section 2, such formation can neither be detected by IPR due to the large sphere radii obtained from the more scattered $R$ points nor GeomCA due to the sparsification removing majority of dense $E$ points that are concatenated to the $R$ cluster, while a single score of GS is too uninformative. In Appendix B.2, we provide a detailed investigation where we both analyzed the lengths of edges in $\mathrm{MST}(\mathcal{G}_D)$ and applied DCA to $R$ and $E$ sets obtained at different truncation levels. The obtained results verify the irregular formation of the points and highlight ineffectiveness of existing methods in such scenarios.

Lastly, note that network consistency $c$ and quality $q$ values are reversed in DCA compared to GeomCA as we consider all 50000 points. Therefore, we trivially obtained $c$ equal to 1 that is perfectly aligned with $q^{sG}$ score due to the geometric sparsification (see Appendix B). On contrary, $q$ increased with $\psi$ except for the observed irregularity at $\psi = 0.5$.

**Quality of individual generated images** A challenging problem in generative modelling is to assess the quality of individual generated samples, which is possible with our q-DCA extension by analyzing the shortest edges of their corresponding representations added to an existing $\mathcal{G}$.

To demonstrate this, we again created $R$ from representations of randomly sampled 50000 training images, approximated $\mathcal{G} = \mathcal{G}_{DD}(R)$ and analyzed newly added edges $\mathcal{E}^{N(q)}$ for each query representation $q$ corresponding to 1000 generated images. In Figure 7, we visualize examples of generated images with shortest (left) and largest (right) inserted edges. We observe that the length of the inserted edges correlates well with the visual quality of generated images, where generated images close to the training ones display clear faces (left) and those further away display distorted ones (right). We compared our results with the realism score (RS) introduced by Kynkäänniemi et al. (2019) and sorted the generated images both by increasing Delaunay edge length and decreasing RS and computed the intersection of first and last 100 samples. We obtained that 51% and 83% of images are commonly marked as high and low quality, respectively, by both methods. Following the discussion in Section 2, we emphasize that RS is highly affected both by outliers and uneven distribution of points in $R$, which the authors address by discarding half of the hyperspheres with the largest radii. Such adjustments are not robust to various arrangements or small perturbations of points, and are not needed in case of Delaunay graphs.

**Additional ablation study on DCA hyperparameters** We use the high dimensionality of representations in this experiment to perform an ablation study on the hyperparameters of our proposed DCA method. Similarly as for the experimental setup in Section 4.1, we investigated the stability of our scores with respect to variations in *i)* the number of sampled rays $T$ in the approximation of the

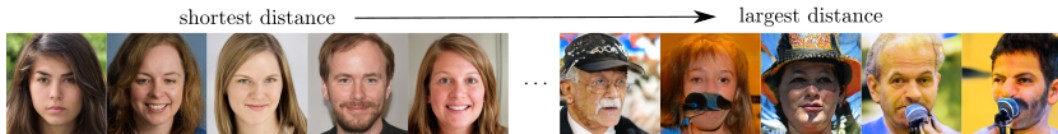

shortest distance ⟶ largest distance

Figure 7: Examples of images generated by a StyleGAN that are closest (left) and furthest (right) to training images from FFHQ dataset in representation space of a pretrained VGG16.

Delaunay graph $\mathcal{G}_D$ (Phase 1), *ii)* the minimum cluster size $mcs$ parameter used in the distillation of $\mathcal{G}_D$ (Phase 2) and lastly, *iii)* the optional sphere coverage parameter $B$ that can be used to reduce the number of edges in the distilled Delaunay graph $\mathcal{G}_{DD}$. The results, shown in Table 8, show that our DCA framework is stable with respect to the choice of hyperparameters even in higher dimensional representation spaces. We refer the reader to Appendix B.2 for further details.

## 4.3    REPRESENTATION SPACE OF VGG16

Lastly, we applied q-DCA to analyze whether the structure of the representation space of a VGG16 classification model pretrained on ImageNet reflects the human labeling. By repeating the experiment investigating separability of classes performed by Poklukar et al. (2021), we reached a similar conclusion that VGG16 moderately separates semantic content (see Appendix B.3). However, we further investigated whether the limited class separation originates from inadequacies of the model or human labeling inconsistencies. To distinguish between the two scenarios, we applied q-DCA to obtain labels of query points and compared them to their true labels.

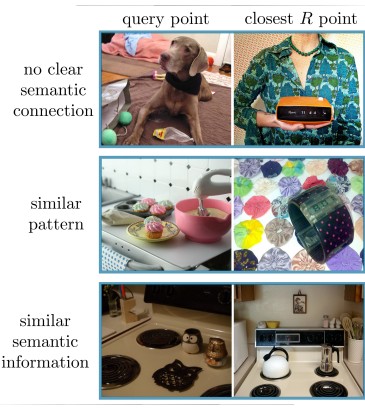

query point    closest $R$ point

no clear semantic connection

similar pattern

similar semantic information

Figure 8: Examples of query images (left) and their corresponding closest R images (right).

**Amending labelling inconsistencies** We used representations of ImageNet images constructed by Poklukar et al. (2021) corresponding to 5 classes representing kitchen utilities and 5 classes representing dogs. We constructed $R$ by randomly sampling 10000 points, and composed the query set $Q_1$ of the remaining 2758 points. We then determined the label of each query point $q$ by the label of its closest point in its neighbourhood $N(q)$. The obtained 87.6% accuracy offers interesting insights into both human labeling and structure of the VGG16 representation space. In Figure 8, we show various examples of query images (left) and the corresponding closest image in $\mathcal{G}_{DD}$ (right) *having different labels*. The first row shows examples of pairs of images considered close by VGG16 despite no clear semantic connection between them. The second row shows examples having meaningfully different labels but are encoded close by VGG16 potentially due to the similar pattern. On the other hand, the last row shows examples where human labeling is different but VGG16 correctly encoded the images close due to their striking semantic similarity. More examples can be found in Appendix B.3, where we show similar cases using images corresponding to randomly chosen classes also considered by Poklukar et al. (2021). The results suggest that a more reliable labeling could result from a combination of human input and the ability of neural networks to recognize semantic similarity in images.

## 5    CONCLUSION

We presented Delaunay Component Analysis (DCA) framework for evaluation of learned data representations which compares the structure of two sets of representations $R$ and $E$ using the introduced distilled Delaunay graphs. We showed that distilled Delaunay graphs provide a reliable approximation in various complex geometric arrangements of $R$ and $E$, thus resulting in a stable output of our DCA scores. Moreover, we introduced the extended q-DCA framework for individual query representations evaluation. We experimentally demonstrated the applicability and flexibility of q-DCA on three different scenarios.

## ACKNOWLEDGEMENTS

This work has been supported by the Knut and Alice Wallenberg Foundation, Swedish Research Council and European Research Council.

## REPRODUCIBILITY STATEMENT

Full implementation of our DCA algorithm together with the code for reproducing our experiments is available on our GitHub[4]. We provide datasets used in all experiments in Section 4.1 and the q-DCA experiment in Section 4.2. For all other experiments, we provide the code containing the processing steps that we applied on datasets provided by Poklukar et al. (2021) (see files named `*_utils.py`). The full description of hyperparameters used in our experiments is available in Appendix B. Lastly, in Appendix A we provide additional explanation of our DCA method presented in Section 3.

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

## A  METHOD: ADDITIONAL BACKGROUND

Our DCA method utilizes several existing methods in each of the three phases, namely, *i)* Delaunay approximation algorithm (Polianskii & Pokorny, 2019) for obtaining the approximated Delaunay graph $\mathcal{G}_D$ built on the given set of representations (Phase 1), *ii)* HDBSCAN (McInnes et al., 2017) for obtaining the distilled Delaunay graph $\mathcal{G}_{DD}$ from $\mathcal{G}_D$ (Phase 2), and lastly *iii)* GeomCA evaluation scores (Poklukar et al., 2021) for analyzing the connected components of $\mathcal{G}_{DD}$ (Phase 3). Therefore, DCA naturally inherits hyperparameters associated to the methods used in Phases 1 and 2, which we summarize below.

---

**Algorithm 1** Delaunay Component Analysis (DCA)

---

**Require:** sets of representations $R$ and $E$
**Optional input:**   number of sampled rays $T$ (default $10^4$)
**Optional input:**   sphere coverage parameter $B$ (default 1.0)
**Optional input:**   minimum cluster size $mcs$ (default 10)
**Optional input:**   component consistency threshold $\eta_c$ (default 0.0)
**Optional input:**   component quality threshold $\eta_q$ (default 0.0)
  **[Phase 1: Manifold approximation]**
  $\mathcal{G}_D \leftarrow$ approximate_Delaunay_graph$(R, E, T)$
  **if** $B < 1.0$ **then**
    $\mathcal{G}_D \leftarrow$ filter_Delaunay_graph$(\mathcal{G}_D, B)$ (Section 3.2)
  **end if**
  **[Phase 2: Component distillation]**
  $\mathcal{G}_{DD} \leftarrow$ distil_Delaunay_graph$(\mathcal{G}_D, mcs)$
  **[Phase 3: Component evaluation]**
  $\mathcal{C} \leftarrow$ get_connected_components$(\mathcal{G}_{DD})$
  $\mathcal{S}_{\text{local}} \leftarrow$ zeros$(|\mathcal{C}|, 2)$
  **for** $i = 0, \ldots, |\mathcal{C}|$ **do**
    $\mathcal{G}_i \leftarrow \mathcal{C}[i]$
    compute $c(\mathcal{G}_i)$ and $q(\mathcal{G}_i)$ as in Definition 3.2 (local scores)
    $\mathcal{S}_{\text{local}}[i, :] \leftarrow [c(\mathcal{G}_i), q(\mathcal{G}_i)]$
  **end for**
  compute $c(\mathcal{G}_{DD}), q(\mathcal{G}_{DD}), \mathcal{P}(\eta_c, \eta_q), \mathcal{R}(\eta_c, \eta_q)$ as in Definition 3.3 (global scores)
  $\mathcal{S}_{\text{global}} \leftarrow [c(\mathcal{G}_{DD}), q(\mathcal{G}_{DD}), \mathcal{P}(\eta_c, \eta_q), \mathcal{R}(\eta_c, \eta_q)]$
**Return:**  $\mathcal{S}_{\text{local}}, \mathcal{S}_{\text{global}}$

---

**Delaunay Graph Approximation** As discussed in Section 3, the Monte Carlo based algorithm presented by Polianskii & Pokorny (2019) samples rays originating from each representation $z \in R \cup E$ and efficiently finds their intersection with the boundary of the Voronoi cell $\text{Cell}(z)$. The points on the boundary then correspond to edges in the Delaunay graph $\mathcal{G}_D$ by Definition 3.1, allowing to iteratively reconstruct $\mathcal{G}_D$ via subgraph approximation. Consequently, the number of found edges directly depends on the number $T$ of rays sampled from each $z$. However, as we show in ablation studies in Appendix B.1, our obtained distilled Delaunay graph $\mathcal{G}_{DD}$ and DCA scores are stable with respect to variations in $T$. In all our experiments, we used $T = 10^4$ unless the value was decreased for computational purposes.

**HDBSCAN** is a hierarchical clustering algorithm that extracts flat clusters using a technique determining the stability of clusters. To reduce the sensitivity to outliers and noise, HDBSCAN first performs a transformation of space using which points with low density are spread apart. Formally, this is achieved by computing *mutual reachability distance* among each pair of points from a given set $W$ defined as $d_{\text{mr-k}}(z_i, z_j) = \{d_k(z_i), d_k(z_j), d(z_i, z_j)\}$, where $z_i \neq z_j \in W$ and $d_k(z_i)$ is the distance to $k$th NN of $z_i$. This induces the first so-called minimum samples hyperparameter of

**Algorithm 2** query point Delaunay Component Analysis (q-DCA)

---

**Require:** Delaunay graph $\mathcal{G}_{DD}(R)$ built on a set of representations $R$
**Require:** set of query representations $Q$
**Optional input:** number of sampled rays $T$ (default $10^4$)
**Optional input:** sphere coverage parameter $B$ (default 1.0)
  $\mathcal{S}_{\text{query}} \leftarrow \texttt{zeros}(|Q|, 3)$
  **for** $j = 0, \dots, |Q|$ **do**
    $q \leftarrow Q[j]$
    $N(q) = (\mathcal{V}^{N(q)}, \mathcal{E}^{N(q)}) \leftarrow \texttt{get\_Delaunay\_neighbourhood\_graph}(\mathcal{G}_{DD}(R), q, T)$
    **if** $B < 1.0$ **then**
      $N(q) \leftarrow \texttt{filter\_Delaunay\_neighbourhood\_graph}(N(q), B)$ (Section 3.2)
    **end if**
    [Processing option 1: Section 4.3]
    $z_q \leftarrow \arg\min_{z_i \in \mathcal{V}^{N(q)} \cap R} d(z_i, q)$    [get closest representation in $R$]
    [Processing option 2: Section 4.2]
    $l_q \leftarrow d(z_q, q)$    [get length of the edge to the closest representation in $R$]
    [Processing option 3: Section 4.1]
    $\hat{\mathcal{E}}^{N(q)} \leftarrow \{\}$    [initiate the set of typical edges]
    **for** $\mathcal{G}_f \in \mathcal{F}$ **do**
      calculate $\mu(\mathcal{E}^{\mathcal{G}_f}), \sigma(\mathcal{E}^{\mathcal{G}_f})$ of edge lengths in $\mathcal{E}^{\mathcal{G}_f}$
    **end for**
    **for** $(z_i, q) \in \mathcal{E}^{N(q)}$ **do**
      **if** $d(z_i, q) \leq \mu(\mathcal{E}^{\mathcal{G}_f}) + \sigma(\mathcal{E}^{\mathcal{G}_f})$ for any $\mathcal{G}_f$ **then**
        $\hat{\mathcal{E}}^{N(q)} \leftarrow \hat{\mathcal{E}}^{N(q)} \cup \{(z_i, q)\}$
      **end if**
    **end for**
    $\mathcal{S}_{\text{query}}[i, :] \leftarrow [z_q, l_q, \hat{\mathcal{E}}^{N(q)}]$
  **end for**
**Return:** $\mathcal{S}_{\text{query}}$

---

HDBSCAN, $k$, which we eliminate from DCA by bypassing the computation of $d_{\text{mr-k}}$ as discussed in Section 3. This is possible because the approximated Delaunay graph $\mathcal{G}_D$ obtained in Phase 1 already captures the natural neighbourhood of points in $W$, where edges originating from outliers and noise differ in length and solid angle from edges originating from points in dense and regular regions. Thus, while HDBSCAN obtains the minimum spanning tree MST from the mutual reachability distance matrix, we obtain it directly from $\mathcal{G}_D$. From MST, HDBSCAN then extracts the dendrogram representing the hierarchy of connected components, which is further simplified into a condensed tree. The latter is obtained by categorizing each split in the dendrogram either as a real split, if there is more than minimum cluster size $mcs$ points in each branch, or as outlying points otherwise. Lastly, by analyzing the distances associated to the birth and death of the clusters in the condensed tree, HDBSCAN extracts flat clusters with longest lifetime. Therefore, with our modification of HDBSCAN, DCA inherits only one hyperparameter $mcs$, which is intuitive to tune. In Appendix B.1, we show that our obtained distilled Delaunay graph $\mathcal{G}_{DD}$ and DCA scores are stable with respect to the choice of $mcs$. In all our experiments we used $mcs = 10$.

We summarize our DCA framework in Algorithm 1 and its q-DCA extension in Algorithm 2.

## B   ADDITIONAL EXPERIMENTAL RESULTS

In this section, we provide further results supporting the experiments discussed in Section 4 as well as present results of our additional experiments. Moreover, we report the hyperparameters used for DCA and the benchmark methods GS, IPR and GeomCA in each experiment.

**Hyperparameters** In Table 1, we report the hyperparameters of all the methods used in our experiments presented in Section 4 and Appendix B. For IPR, we used neighborhood size $k = 3$ as recommended by the authors and constructed the sets $R$ and $E$ of equal size by randomly sampling $\min(|R|, |E|)$ points from each of them. For GS and GeomCA, we followed the same hyperparame-

ter choices as in (Poklukar et al., 2021). By default, we used GeomCA with geometric sparsification as suggested by Poklukar et al. (2021) unless specified otherwise. Note that using $\delta = \epsilon$ trivially results in perfect network quality $q$ as it necessarily yields only heterogeneous edges among points from $R$ and $E$ (see (Poklukar et al., 2021) for details). In some cases, we also applied GS on the sets $R$ and $E$ of equal size which were constructed as for IPR. For DCA, we set $mcs = 10$ for all experiments. Whenever computations were feasible on a single GPU, we used $T = 10^4$ and performed no filtering of $\mathcal{G}_D$ (i.e., used $B = 1.0$). This was possible when evaluating contrastive representations in Section 4.1 and representations obtained from VGG16 in Section 4.3. When evaluating generation capabilities of a StyleGAN in Section 4.2, we instead sampled less rays $T$ in the approximation of the Delaunay graph $\mathcal{G}_D$ and used $B = 0.7$.

Table 1: Hyperparameters used for the benchmark methods GS, IPR and GeomCA as well as our DCA method in the experiments presented in Section 4. The same hyperparameter choices apply for experiments presented in Appendix B.

| | | GS | | | IPR | GeomCA | | DCA | | |
|---|---|---|---|---|---|---|---|---|---|---|
| | $L_0$ | $\gamma$ | $i_{max}$ | $n$ | $k$ | $\varepsilon(p)$ | $\delta$ | $T$ | $B$ | $mcs$ |
| Section 4.1 | 64 | 1/128 | 10 | 1000 | 3 | $\varepsilon(1)$ | $\varepsilon/2$ | $1 \cdot 10^4$ | 1.0 | 10 |
| Section 4.2 | 64 | 1/128 | 100 | 1000 | 3 | $\varepsilon(10)$ | $\varepsilon$ | $5 \cdot 10^3$ | 0.7 | 10 |
| Section 4.3 | / | / | / | / | 3 | $\varepsilon(10)$ | $\varepsilon$ | $1 \cdot 10^4$ | 1.0 | 10 |

## B.1 Contrastive Representations

In this section, we first derive the optimal values of precision and recall scores used in the *varying component density* experiment in Section 4.1 and present GS score results for the same experiment. We then show how q-DCA can be applied to investigate the robustness of the model's representation space to novel images, and present the results of the additional *mode truncation* experiment performed in (Poklukar et al., 2021). Lastly, we present the results of an extensive ablation study on the inherited hyperparameters of DCA: $T$, $mcs$ and the optional $B$ parameter, described in Appendix A.

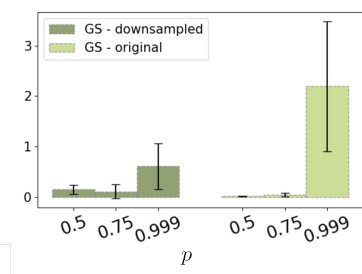

Figure 9: Mean and standard deviation of GS scores (multiplied by 100) obtained over 10 iterations when varying the fraction $p$ controlling the amount of discarded $R$ points. Left: GS calculated on $R$ and $E$ of equal size by downsampling $E$. Right: GS calculated on the original $R$ and $E$ sets of unequal size.

**Varying component density** In Table 3, we report the number of representations corresponding to each class in the considered training $\mathcal{D}_f^R$ and test $\mathcal{D}_f^E$ datasets containing images of boxes placed in 12 possible configurations recorded from the front camera view. The total number of representations corresponding to the first 7 classes $c_0, \dots, c_6$ contained in $R$ and $E$ was 3514 and 3463, respectively. Therefore, on average, each class in $R$ contained 502 points. By Definition 3.3, recall $\mathcal{R}$ is defined as the fraction of $R$ points contained in the union of fundamental components $\mathcal{F}$ over the total number of points in $R$. Thus, when discarding $p = 0.5$ fraction of points from three fundamental components in $R$, the average optimal value of recall $\mathcal{R}_*$ is calculated as $\mathcal{R}_* \approx (502 \cdot 4)/(502 \cdot 4 + 251 \cdot 3) \approx 0.73$. When $p = 0.75$, we get that $\mathcal{R}_* \approx (502 \cdot 4)/(502 \cdot 4 + 126 \cdot 3) \approx 0.84$, and for $p = 0.999$, $\mathcal{R}_* \approx (502 \cdot 4)/(502 \cdot 4 + 1 \cdot 3) \approx 0.99$. Similarly, since precision $\mathcal{P}$ is defined as the fraction of $E$ points contained in $\mathcal{F}$ over the total number of points in $E$, its average optimal value is calculated as $\mathcal{P}_* \approx (494 \cdot 4)/3463 \approx 0.57$. This value is optimal for all values of $p$ since variations in $p$ do not affect the number of points in $E$.

In Figure 9, we additionally report the mean and standard deviation (std) of GS scores (multiplied by 100) obtained over 10 iterations as in Section 4.1 when varying $p \in \{0.5, 0.75, 0.999\}$. We calculated GS both on $R$ and $E$ sets of equal size by downsampling the set $E$ as in calculation of IPR (dark green) and original $R$ and $E$ of unequal sizes (light green) as used in DCA. Firstly, we see that the results are not robust to the changes in the number of points in each set (dark vs light green) and can exhibit large std which is most evident in the case of

outliers for $p = 0.999$. The results show that GS does not reflect the geometric arrangement of the $R$ and $E$ points which illustrates inadequate informativeness of the score.

**Cluster assignments** We applied our q-DCA algorithm, presented in Section 3.1, to analyze the robustness of the model to novel images not seen during the training. For this, we constructed three different query sets: $Q_1$ containing 100 test representations of classes $c_0, \ldots, c_6$ that were removed from $E$, $Q_2$ containing 100 test representations of classes $c_7, \ldots, c_{11}$, and $Q_3$ comprising 250 randomly chosen representations of images of all classes that were recorded from the right camera view (see Figure 4, right). As $R$ and $E$ contain only representations of classes $c_0, \ldots, c_6$, we expected $Q_1$ to be close to existing fundamental connected components in the distiled Delaunay graph $\mathcal{G} = \mathcal{G}_{DD}(R \cup E)$ and $Q_2$ further away. Moreover, if the model is robust to the semantic content in the right-view images, points in $Q_3$ labeled with first 7 classes were expected to be close to $\mathcal{G}$, while the remaining ones should lie further apart.

To obtain an assignment (if any) of a query point $q_j \in Q_k$ to an existing fundamental component $\mathcal{G}_f = (\mathcal{V}^{\mathcal{G}_f}, \mathcal{E}^{\mathcal{G}_f}) \in \mathcal{F}$, we analyzed the edges inserted using q-DCA (see also Algorithm 2 in Appendix A). Recall that due to the nature of Delaunay graphs, $q_j$ can have edges associated to several $\mathcal{G}_f$ although these vary in their length. Therefore, we considered only those inserted edges whose length were representative of the connected components they connected to, i.e, we calculated the mean $\mu(\mathcal{E}^{\mathcal{G}_f})$ and std $\sigma(\mathcal{E}^{\mathcal{G}_f})$ of the length of edges in each $\mathcal{G}_f \in \mathcal{F}$ and kept only edges from $N(q_j) \cap \mathcal{G}_f$ that were shorter than $\mu(\mathcal{E}^{\mathcal{G}_f}) + \sigma(\mathcal{E}^{\mathcal{G}_f})$. We refer to these edges as *typical edges* and define two assignment procedures: (1) *conservative* where we assigned $q_j$ to a component $\mathcal{G}_f$ only if all typical edges connected $q_j$ to exactly one $\mathcal{G}_f$ and did not assign it to any otherwise, and (2) *flexible* where we additionally assigned $q_j$, whose typical edges connected to more than one component, to a component $\mathcal{G}_f$ if it attained both shortest and maximum number of typical edges among all the candidate components.

Using class labels $c_i$, we first determined the majority-vote label of each $\mathcal{G}_f$ which we observed to be $> 99.9\%$ accurate (see also Table 4 in Appendix B.1). Given the labels of $q_j \in Q_k$, we then extracted those $q_j$ that *a)* were assigned to a component and report the percentage $A$ of those that were assigned to the one with the correct label. However, by design, our assignment procedure might not assign all the points. Therefore, we additionally extract $q_j$ that *b)* should be assigned to a component and report the percentage $B$ of them that were correctly assigned. Thus, for $Q_1$, the extracted representations in *b)* is the entire set $Q_1$. Same holds for $Q_2$ as in this case we assumed that points in $Q_2$ should not be assigned to any component and thus report the percentage of those that were correctly not assigned to any. In $Q_3$, we consider a combination of the two cases, where 1750 representations corresponding to $c_0, \ldots, c_6$ should be assigned to a component if the model is able to extract the underlying semantic information, while the remaining 1250 representations corresponding to $c_i, i \geq 7$ should not. Therefore, for $A$, the higher the better for $Q_1$, the lower the better for $Q_2$ and $A \approx 1750/3000$ for $Q_3$, while for $B$ it holds that the higher the better for all $Q_i$.

Table 2: Results of assigning points in query sets $Q_k$ to fundamental components $\mathcal{G}_f \in \mathcal{F}$.

| | $A = \%$ CORRECTLY ASSIGNED | | | $B = \%$ THAT SHOULD BE ASSIGNED | | |
| | $Q_1(\uparrow)$ | $Q_2(\downarrow)$ | $Q_3$ | $Q_1(\uparrow)$ | $Q_2(\uparrow)$ | $Q_3(\uparrow)$ |
|---|---|---|---|---|---|---|
| CONSERVATIVE | 0.99 | 0.00 | 0.08 | 0.68 | 0.85 | 0.28 |
| FLEXIBLE | 0.99 | 0.00 | 0.08 | 0.98 | 0.83 | 0.19 |

We show the assignment results in Table 2. For $Q_1$, we see that the conservative approach resulted in $A = 99\%$ and $B = 68\%$. The former is because 3 points were wrongly assigned, while the latter is due to the fact that 219 representations out of 700 were assigned to several components. Out of these, 206 were considered in the flexible assignment yielding an increase of $B$ to $98\%$ of correct assignments. For $Q_2$, the conservative approach resulted in $B = 85\%$ due to 77 points being assigned to a component despite their wrong label $c_i, i \geq 7$ (thus leading to $A = 0\%$), suggesting a moderate separation of classes. For $Q_3$, we see that $A = 8\%$ and $B = 28\%$ in the conservative approach which indicates that the model was not able to recognize similar box configurations recorded from different camera view, which is aligned with the results obtained by Chamzas et al. (2021). Lastly, we observe that flexible approach yielded only minor decrease in $B$ for $Q_2$ and $Q_3$.

Table 3: Number of representations corresponding to each class $c_i$ contained in the training $\mathcal{D}_f^R$ (top row) and test $\mathcal{D}_f^E$ (middle row) datasets. The bottom row shows the size of the sets $E_t$ used in the mode truncation experiment.

| $c_i$ | 0 | 1 | 2 | 3 | 4 | 5 | 6 | 7 | 8 | 9 | 10 | 11 |
|---|---|---|---|---|---|---|---|---|---|---|---|---|
| $\mathcal{D}_f^R$ | 670 | 690 | 395 | 706 | 349 | 409 | 295 | 296 | 292 | 311 | 258 | 331 |
| $\mathcal{D}_f^E$ | 666 | 625 | 373 | 684 | 429 | 377 | 309 | 312 | 310 | 293 | 279 | 345 |
| $E_t$ | 666 | 1291 | 1664 | 2348 | 2777 | 3154 | 3463 | 3775 | 4085 | 4378 | 4657 | 5002 |

Figure 10: Scores obtained on $R$ and $E_t$ when varying the truncation parameter $t$. Left: DCA $\mathcal{P}$, GeomCA with sparsification $\mathcal{P}^{sG}$, and IPR $\mathcal{P}^I$ precision scores (blue) together with GS score multiplied by 100 (green). Middle: DCA $\mathcal{R}$, GeomCA with sparsification $\mathcal{R}^{sG}$, and IPR $\mathcal{R}^I$ recall scores. Right: network consistency (green) and quality (purple) scores of DCA ($c, q$) and GeomCA with sparsification ($c^{sG}, q^{sG}$).

**Mode Truncation** We repeated the mode truncation experiment performed by Poklukar et al. (2021) and applied DCA on the set $R$ comprised of representations corresponding to images of the first 7 classes, $c_0, \ldots, c_6$, and the sets $E_t$ containing representations corresponding to images of the first $t$ classes $c_0, \ldots, c_t$ for $t \leq 11$ (see Table 3 for the exact number of points in each of these sets). Since we expect representations to be separated by the class label due to the contrastive training objective, we again set $\eta_c, \eta_q$ defining fundamental components to high values of $0.75$ and $0.45$, respectively. By design, the sets $R$ and $E_t$ should reflect mode collapse scenario for $t < 6$ and mode discovery scenario for $t > 7$, while in the case of $R$ and $E_6$ we expect to observe a perfect alignment with exactly 7 fundamental connected components.

Figure 10 shows the obtained $\mathcal{P}, \mathcal{R}$ (left) and $c, q$ (right) DCA scores for each of the sets $R \cup E_t$ for $t \in \{0, \ldots, 11\}$. We observe that DCA correctly correlated with the number of (dis)covered modes contained in $E_t$. The precision $\mathcal{P}$ (solid blue) increased with $t$, while recall $\mathcal{R}$ (solid orange) decreased. Moreover, the highest scores were achieved at $t = 6$ where all the modes in $R$ were covered in $E_6$. The IPR scores reflect the same behaviour but resulted in lower absolute values. This is likely due to the small radius of the constructed hyperspheres arising from a small distance between representations of the same class (as discussed in Section 2). We observe that GS (multiplied by 100) failed to correctly reflect mode collapse for $t \in \{0, 1\}$ as well as mode discovery cases, which illustrates that GS is not sufficiently informative. Additionally, we see that precision and recall (left), and component consistency and quality (right) are aligned for both DCA ($\mathcal{P}, \mathcal{R}, c, q$ marked with solid lines) and GeomCA ($\mathcal{P}^{sG}, \mathcal{R}^{sG}, c^{sG}, q^{sG}$ marked with dashed lines). This is because the similar density of connected components in this case yielded a reliable $\varepsilon$ estimation (see Section 2). Lastly, for DCA, network consistency $c$ (green) and quality $q$ (purple) correctly increased until they reached the maximum at $t = 6$ and then started to decrease. Note that, in contrast to GeomCA, DCA utilized all the points.

**Ablation Study** We present the results of an ablation study on DCA hyperparameters discussed in Appendix A and Section 3.2, and show that our algorithm is robust to their choices. We performed an ablation study on *Ab-1)* HDBSCAN min cluster size $mcs$ parameter, *Ab-2)* sphere coverage parameter $B$ and *Ab-3)* the parameter $T$ determining the number of rays used in the Delaunay graph approximation algorithm. To ensure a controlled representation space where prior knowledge of its structure is available, we used the sets $R$ and $E_6$ as in previous experiments. However, to fairly evaluate the obtained scores, we additionally balanced the sets to contain 250 representations corresponding to each of the 7 classes. Thus, $|R| = |E_6| = 1750$ which always yielded a perfect network consistency score $c$ and is therefore omitted from the results. For all experiments, we set $\eta_c = 0.75$ and $\eta_q = 0.45$ in order to anchor the analysis on the fundamental components only. We always fixed $T = 10^4$ in the approximation of $\mathcal{G}_D$, $B = 1.0$ (i.e. preformed no filtering of $\mathcal{G}_D$) and $msc = 10$ for HDBSCAN except in *Ab-1)* where we varied $msc \in \{3, 5, 10, 20\}$, in *Ab-2)* where we varied $B \in \{0.5, 0.6, 0.7, 0.8, 0.9, 1.0\}$, and in *Ab-3)* where we varied $T \in \{10, 10^2, 10^3, 10^5, 10^6\}$.

VALIDITY OF FUNDAMENTAL COMPONENTS: Firstly, for each experiment in *Ab-i)*, we verified that the fundamental components in the distilled Delaunay graph $\mathcal{G}_{DD}$ contain representations corresponding to the same class. In Table 4, we report the percentage of points in $R \cup E_6$ that contributed to the majority-vote label of each fundamental connected component $\mathcal{G}_f \in \mathcal{F}$ (top rows) when varying $mcs$ (columns) in *Ab-1)*. In square brackets, we report the obtained majority-vote label $[c_i]$ corresponding to each $\mathcal{G}_f$. In bottom rows, we report the total percentage of points that contributed to the majority-vote labelling (row *included*) as well as the percentage of points that were not contained in any fundamental component $\mathcal{G}_f$ (row *excluded*). Firstly, for any choice of $mcs$, the majority-vote labels of the fundamental components are unique and include all classes $c_0, \ldots, c_6$. Secondly, we observe that only $mcs = 3$ resulted in $|\mathcal{F}| > 7$, where points of class $c_3$ were split in two fundamental components. Thirdly, for any choice of $mcs$, each $\mathcal{G}_f$ correctly contains approximately $1/7 \approx 0.14\%$ of all the 3500 points in $R \cup E_6$, except for $f = 6$ (and $f = 10$) where a small fraction of points were excluded. However, we see that with increasing $mcs$, the percentage of excluded points decreased and stabilized for $mcs \geq 10$. In *Ab-2)* and *Ab-3)*, we observed the same values as for $mcs \geq 10$ in Table 4. These results show that DCA is robust to the choice of hyperparameters, resulting in the correct distilled Delaunay graph $\mathcal{G}_{DD}$.

Table 4: The percentage of points in $R \cup E_6$ contributing to the majority-vote labelling of each fundamental component $\mathcal{G}_f \in \mathcal{F}(\mathcal{G}_{DD}(R \cup E_6), \eta_c = 0.75, \eta_q = 0.45)$ (shown in rows) when varying $mcs$ (columns). The majority vote label $c_i$ of each $\mathcal{G}_f$ is shown in square brackets $[c_i]$. The bottom rows show the total percentage of points from $R \cup E_6$ that contributed to the majority-vote labelling (row *included*) and that were excluded from fundamental components $\mathcal{F}$ (row *excluded*).

| | | % OF POINTS LABELLED WITH MAJORITY-VOTE $[c_i]$ | | | |
|---|---|---|---|---|---|
| $mcs$ | | 3 | 5 | 10 | 20 |
| | 0 | $0.143\ [c_2]$ | $0.143\ [c_2]$ | $0.143\ [c_2]$ | $0.143\ [c_2]$ |
| | 1 | $0.142\ [c_4]$ | $0.142\ [c_1]$ | $0.142\ [c_1]$ | $0.142\ [c_1]$ |
| | 2 | $0.141\ [c_5]$ | $0.142\ [c_4]$ | $0.142\ [c_4]$ | $0.142\ [c_4]$ |
| $\mathcal{G}_f$ | 3 | $0.140\ [c_1]$ | $0.141\ [c_5]$ | $0.141\ [c_5]$ | $0.141\ [c_5]$ |
| COMP. | 4 | $0.139\ [c_0]$ | $0.140\ [c_6]$ | $0.140\ [c_6]$ | $0.140\ [c_6]$ |
| IDX | 5 | $0.127\ [c_6]$ | $0.139\ [c_0]$ | $0.139\ [c_0]$ | $0.139\ [c_0]$ |
| | 6 | $0.127\ [c_3]$ | $0.127\ [c_3]$ | $0.136\ [c_3]$ | $0.136\ [c_3]$ |
| | 10 | $0.001\ [c_3]$ | / | / | / |
| INCLUDED [%] | | 0.960 | 0.973 | 0.981 | 0.981 |
| EXCLUDED [%] | | 0.040 | 0.027 | 0.018 | 0.018 |

STABILITY OF DCA SCORES: Next, in each *Ab-i)*, we measured the stability of our precision $\mathcal{P}$, recall $\mathcal{R}$, network quality $q$, as well as the number of fundamental components $|\mathcal{F}|$ with respect to the varying hyperparameter. Moreover, since variations in all hyperparameters affect the number of edges obtained in the resulting distilled Delaunay graph $\mathcal{G}_{DD}$, we also report percentage of edges normalized by the highest number obtained in each ablation experiment separately. Note that varying $T$ directly affects the number of found edges in $\mathcal{G}_D$, while varying $mcs$ affects the number of edges in $\mathcal{G}_{DD}$ because of the definition of outlying points by HDBSCAN. The results are shown in Table 5, where the column % *edges* in square brackets additionally shows the maximum number of edges

Table 5: DCA ablation results obtained when varying $mcs$ in experiment *Ab-1)* (top rows), sphere coverage parameter $B$ in *Ab-2)* (middle rows) and number of sampled rays $T$ in *Ab-3)* (bottom rows). For each experiment *Ab-i)*, we report in columns the obtained precision $\mathcal{P}$, recall $\mathcal{R}$, network quality $q(\mathcal{G})$, percentage of edges normalized by the maximum number obtained per experiment *Ab-i)* (with maximum absolute value shown in square brackets) and lastly, the number of fundamental components $|\mathcal{F}|$.

| | | $\mathcal{P}$ | $\mathcal{R}$ | $q(\mathcal{G})$ | % EDGES | $|\mathcal{F}|$ |
|---|---|---|---|---|---|---|
| $mcs$ | 3 | 0.949 | 0.972 | 0.502 | 0.959 | 8 |
| | 5 | 0.967 | 0.979 | 0.502 | 0.984 | 7 |
| | 10 | 0.977 | 0.987 | 0.502 | 1.0 | 7 |
| | 20 | 0.977 | 0.987 | 0.502 | 1.0 [301376] | 7 |
| | | [0.967 ± 0.013] | [0.981 ± 0.007] | [0.502 ± 0.000] | | |
| $B$ | 0.5 | 0.977 | 0.989 | 0.502 | 0.357 | 7 |
| | 0.6 | 0.977 | 0.987 | 0.502 | 0.393 | 7 |
| | 0.7 | 0.977 | 0.987 | 0.502 | 0.448 | 7 |
| | 0.8 | 0.977 | 0.987 | 0.502 | 0.534 | 7 |
| | 0.9 | 0.977 | 0.987 | 0.502 | 0.676 | 7 |
| | 1.0 | 0.977 | 0.987 | 0.502 | 1.0 [301376] | 7 |
| | | [0.977 ± 0.000] | [0.988 ± 0.00] | [0.502 ± 0.000] | | |
| $T$ | 10 | 0.975 | 0.982 | 0.498 | 0.056 | 7 |
| | $10^2$ | 0.977 | 0.987 | 0.501 | 0.223 | 7 |
| | $10^3$ | 0.977 | 0.987 | 0.503 | 0.467 | 7 |
| | $10^4$ | 0.977 | 0.987 | 0.502 | 0.693 | 7 |
| | $10^5$ | 0.977 | 0.987 | 0.501 | 0.871 | 7 |
| | $10^6$ | 0.977 | 0.987 | 0.501 | 1.0 [434802] | 7 |
| | | [0.976 ± 0.000] | [0.987 ± 0.002] | [0.501 ± 0.002] | | |

obtained in each *Ab-i)* used in the respective normalization. Firstly, we observe that the obtained $\mathcal{P}, \mathcal{R}$ and $q$ are robust to various choices of hyperparameters, with minor deviations for $mcs = 3$ which is also the only case that wrongly resulted in 8 fundamental components. Secondly, in *Ab-1)* (top rows) we observe rather stable number of edges in $\mathcal{G}_{DD}$. In *Ab-2)* (middle rows), we observe that sphere coverage parameter $B$ not only significantly reduces the number of edges in $\mathcal{G}_{DD}$ but also does not affect the obtained scores. This empirically shows that the edge filtering, introduced in Section 3.2, correctly removes unstable edges that are not significant for the correct manifold estimation. We observe a similar result in *Ab-3)* (bottom rows), where increasing the number of sampled rays $T$ naturally increased the number of found Delaunay edges, while again not affecting the resulting scores except for the minor decrease in case of the extremely low $T = 10$. These result show the reliability of both our manifold approximation using Delaunay graphs and the introduced edge filtering using $B$, as well as the stability of our DCA scores.

DCA RUNTIME: Lastly, we additionally report empirical runtime (obtained on NVIDIA GeForce GTX 1650 with Max-Q Design) of the main components of the proposed DCA method normalized by the total elapsed time of each experiment: approximating the Delaunay edges in $\mathcal{G}_D$ (column *approx. $\mathcal{G}_D$*), filtering of Delaunay edges using sphere coverage parameter $B$ (*filter $\mathcal{G}_D$*), distilling $\mathcal{G}_D$ into the distilled Delaunay graph $\mathcal{G}_{DD}$ (*distill $\mathcal{G}_{DD}$*) and the analysis of connected components obtained in $\mathcal{G}_{DD}$ (*analyse $\mathcal{G}_{DD}$*). We also report the total time elapsed in seconds (*total*) for each experiment in *Ab-i)*. The obtained times are shown in Table 6. Firstly, we see that the largest computational bottleneck is the approximation of the Delaunay graph $\mathcal{G}_D$, followed by its filtering using $B$ if performed (middle rows). Secondly, we observe that choosing $B < 1.0$ significantly reduces the number of edges and hence improves the memory consumption (see middle rows in Table 5) at the cost of slightly increasing the total computational time (column *total*). Thirdly, we see that in *Ab-3)* increasing $T$ naturally increases the time necessary for the approximation of $\mathcal{G}_D$ and consequentially the time needed to distill $\mathcal{G}_D$ and analyze $\mathcal{G}_{DD}$. For clarity, we report the actual elapsed time in seconds for columns *distill $\mathcal{G}_D$* and *analyse $\mathcal{G}_{DD}$* because of the potentially misleading percentage obtained from the large normalization constant (column *total*).

Table 6: Empirical runtime of the main components of our DCA algorithm obtained when varying $mcs$ in experiment *Ab-1)* (top rows), sphere coverage parameter $B$ in *Ab-2)* (middle rows) and number of sampled rays $T$ in *Ab-3)* (bottom rows). For each experiment *Ab-i)*, we report the elapsed time normalized by the total elapsed time per experiment of the following implementation parts: approximation of the Delaunay graph $\mathcal{G}_D$ (column *approx. $\mathcal{G}_D$*), filtration of $\mathcal{G}_D$ using sphere coverage parameter $B$ (*filter $\mathcal{G}_D$*), distillation of $\mathcal{G}_D$ into the distilled Delaunay graph $\mathcal{G}_{DD}$ (*distill $\mathcal{G}_{DD}$*) and analysis of connected components obtained in $\mathcal{G}_{DD}$ (*analyse $\mathcal{G}_{DD}$*). For columns *distill $\mathcal{G}_{DD}$* and *analyse $\mathcal{G}_{DD}$* in *Ab-3)*, we additionally report in square brackets the actual elapsed time to alleviate the effects of the large normalization constant. Finally, we report the total time elapsed in seconds (*total*) for each experiment in *Ab-i)*.

|  |  | APPROX. $\mathcal{G}_D$ [%] | FILTER $\mathcal{G}_D$ [%] | DISTIL $\mathcal{G}_D$ [%] | ANALYSE $\mathcal{G}_{DD}$ [%] | TOTAL [$s$] |
|---|---|---|---|---|---|---|
| $mcs$ | 3 | 0.852 | / | 0.046 | 0.029 | 49.6 |
|  | 5 | 0.837 | / | 0.054 | 0.032 | 50.4 |
|  | 10 | 0.841 | / | 0.054 | 0.029 | 50.2 |
|  | 20 | 0.841 | / | 0.051 | 0.031 | 50.2 |
| $B$ | 0.5 | 0.748 | 0.151 | 0.028 | 0.046 | 57.1 |
|  | 0.6 | 0.724 | 0.169 | 0.031 | 0.048 | 59.0 |
|  | 0.7 | 0.732 | 0.155 | 0.033 | 0.050 | 58.4 |
|  | 0.8 | 0.715 | 0.155 | 0.037 | 0.058 | 59.7 |
|  | 0.9 | 0.699 | 0.150 | 0.043 | 0.069 | 61.1 |
|  | 1.0 | 0.729 | / | 0.084 | 0.109 | 58.6 |
| $T$ | 10 | 0.792 | / | 0.085 [0.28$s$] | 0.060 [0.19$s$] | 3.2 |
|  | $10^2$ | 0.622 | / | 0.114 [0.56$s$] | 0.094 [0.46$s$] | 4.9 |
|  | $10^3$ | 0.601 | / | 0.126 [1.50$s$] | 0.084 [1.00$s$] | 12.0 |
|  | $10^4$ | 0.854 | / | 0.043 [2.08$s$] | 0.028 [1.37$s$] | 48.1 |
|  | $10^5$ | 0.975 | / | 0.008 [3.42$s$] | 0.004 [1.80$s$] | 422.6 |
|  | $10^6$ | 0.997 | / | 0.001 [4.39$s$] | 0.001 [2.10$s$] | 4157.8 |

Table 7: The mean and the standard deviation of the lengths of homogeneous edges among points in $R$ and among points in $E$ per truncation level $\psi$.

| $\psi$ | 0.0 | 0.1 | 0.2 | 0.3 | 0.4 | 0.5 |
|---|---|---|---|---|---|---|
| $R$ | $30.07 \pm 6.29$ | $30.11 \pm 6.34$ | $30.22 \pm 6.37$ | $30.35 \pm 6.41$ | $30.48 \pm 6.45$ | $30.47 \pm 6.55$ |
| $E$ | $7.25 \pm 0.85$ | $9.76 \pm 1.44$ | $12.96 \pm 2.50$ | $15.96 \pm 3.29$ | $18.37 \pm 3.78$ | $20.28 \pm 4.09$ |

| $\psi$ | 0.6 | 0.7 | 0.8 | 0.9 | 1.0 |
|---|---|---|---|---|---|
| $R$ | $30.53 \pm 6.62$ | $30.40 \pm 6.59$ | $30.21 \pm 6.66$ | $29.97 \pm 6.61$ | $29.71 \pm 6.53$ |
| $E$ | $21.91 \pm 4.44$ | $23.32 \pm 4.72$ | $24.74 \pm 4.98$ | $26.13 \pm 5.26$ | $27.41 \pm 5.44$ |

## B.2 ADDITIONAL EVALUATION OF STYLEGAN

In this section, we present a detailed investigation of the geometry of $R$ and $E$ points obtained in Section 4.2 as well as an additional ablation study on the hyperparameters of the DCA method similar to the one carried out in Section B.1.

**Investigation of the geometry of $R$ and $E$ points obtained in Section 4.2** Firstly, we additionally verified the correctness of the obtained DCA scores for each truncation level $\psi$ by investigating the average length of homogeneous edges among $R$ and among $E$ points. The results including mean and standard deviation of the lengths are reported in Table 7. We observe that the average length homogeneous edges among $E$ increases with increasing $\psi$ and resembles the average length of the homogeneous edges among $R$ for $\psi = 1.0$. This analysis suggests that the StyleGAN model was successfully trained, which is supported by the high-quality images reported in (Karras et al., 2019), and indicates the correctness of the obtained DCA values.

Next, we performed a series of further experiments to investigate the decrease in our DCA scores obtained at truncation level $\psi = 0.5$ shown in Figure 6. Firstly, we analysed whether the decrease emerged from irregularities of points in $R$ or points in $E$. To this end, we performed two more tests and applied DCA on the sets *a)* $R = R(\psi = 0.4)$ and $E = R(\psi = 0.5)$ and *b)* $R = R(\psi = 0.4)$ and $E = E(\psi = 0.5)$, where we denoted by, e.g., $R(\psi = 0.4)$, the set $R$ obtained at truncation level $\psi = 0.4$ in Section 4.2. In *a)*, we obtained that $\mathcal{P} = 0.98, \mathcal{R} = 0.98$ and $q = 0.55$ indicating no irregularities in the set $R(\psi = 0.5)$ of sampled training image representations, while in *b)* we obtained $\mathcal{P} = 0.52, \mathcal{R} = 0.05$ and $q = 0.14$, which is equally low as for $R = R(\psi = 0.5)$ and $E = E(\psi = 0.5)$ where $\mathcal{P} = 0.65, \mathcal{R} = 0.10$ and $q = 0.19$ (as visualized in Figure 6). This analysis showed that the irregularity originates from the geometry of the set $E(\psi = 0.5)$.

Next, in order to verify that $R$ and $E$ were formed in a specific geometric arragement similar to that depicted in Figure 2, we additionally analyzed the minimum spanning tree $\text{MST}(\mathcal{G}_D)$ obtained from the approximated Delaunay graph $\mathcal{G}_D$ built on the sets $R = R(\psi = 0.5)$ and $E = E(\psi = 0.5)$. Recall that $\text{MST}(\mathcal{G}_D)$ is a subgraph of $\mathcal{G}_D$ connecting all the vertices such that the total edge length is minimized and there exist no cycles. The latter implies that an MST of a graph with $n$ vertices contains $n - 1$ edges. If our hypothesis that $R$ and $E$ have different densities and are concatenated with a minor intersection is correct, this should be reflected in the length of the edges in $\text{MST}(\mathcal{G}_D)$. We calculated the average length and standard deviation of both homogeneous (among points from only one of the sets) and heterogeneous edges (among points from $R$ and $E$) in $\text{MST}(\mathcal{G}_D)$. We obtained the average length of homogeneous edges $21.58 \pm 4.88$ among $R$ and $13.78 \pm 3.29$ among $E$ points, while for heterogeneous edges the length resulted in $18.19 \pm 4.00$. This indicates the differences in the density of points in each of the set. Moreover, homogeneous edges among $R$ and $E$ points represented 33% and 49% of all edges, respectively, while heterogeneous edges accounted only for the remaining 18%. Together with the obtained low network quality $q = 0.19$ showing that $R$ and $E$ are poorly geometrically mixed, this verifies that points from $R$ and $E$ are concatenated as shown in Figure 2. Moreover, when performing geometric sparsification on $R = R(\psi = 0.5)$ and $E = E(\psi = 0.5)$, which resulted in $|R| = 9364$ and $|E| = 576$, we obtained $\mathcal{P} = 0.77, \mathcal{R} = 0.43$ and $q = 0.16$, which artificially improved the DCA scores with a minor decrease in $q$ compared to results obtained in Figure 6. We emphasise that this scenario cannot be detected with the existing methods due to their limitations discussed in Section 2.

**Ablation study** We perform an additional ablation study on DCA hyperparameters discussed in Appendix A and Section 3.2 using high-dimensional representations of training and generated images obtained from the VGG16 model, and show that our algorithm is robust to their choices. As in Section B.1, we performed an ablation study on *Ab-1)* HDBSCAN min cluster size $mcs$ parameter, *Ab-2)* optional sphere coverage parameter $B$ and *Ab-3)* the parameter $T$ determining the number of rays used in the Delaunay graph approximation algorithm. For computational purposes, all the experiments were performed on fixed sets $R$ and $E$ containing 10000 VGG16 representations of training and generated images, respectively. For all experiments, we set $\eta_c = 0.0$ and $\eta_q = 0.0$ as done in Section 4.2. We always fixed $T = 10^4$ in the approximation of $\mathcal{G}_D$, $B = 0.7$ and $msc = 10$ for HDBSCAN except in *Ab-1)* where we varied $msc \in \{3, 5, 10, 20\}$, in *Ab-2)* where we varied $B \in \{0.5, 0.6, 0.7, 0.8, 0.9, 1.0\}$, and in *Ab-3)* where we varied $T \in \{10, 10^2, 10^3, 10^5, 10^6\}$.

We measured the stability of our precision $\mathcal{P}$, recall $\mathcal{R}$, network quality $q$ for each experiment *Ab-i)* with respect to the varying hyperparameter. Since variations in all hyperparameters affect the number of edges obtained in the resulting distilled Delaunay graph $\mathcal{G}_{DD}$, we also report percentage of edges normalized by the highest number obtained in each ablation experiment separately. For each *Ab-i)*, we additionally report the mean and the standard deviation (std) of $\mathcal{P}, \mathcal{R}$ and $q$ obtained when varying the corresponding parameter. Note that varying $T$ directly affects the number of found edges in $\mathcal{G}_D$, while varying $mcs$ affects the number of edges in $\mathcal{G}_{DD}$ because of the definition of outlying points by HDBSCAN.

The results are shown in Table 8, where the column % *edges* in square brackets additionally shows the maximum number of edges obtained in each *Ab-i)* used in the respective normalization. Firstly, we observe that the obtained $\mathcal{P}, \mathcal{R}$ and $q$ are robust to various choices of hyperparameters, except for the $\mathcal{P}, \mathcal{R}$ values obtained in *Ab-1)* when varying $mcs$ parameter (top rows). However, such variations are expected due to the nature of the $mcs$ parameter, which directly affects the obtained connected components. Increasing $mcs$ naturally yields more outliers or components of lower quality, which can be seen in the decrease of the $\mathcal{P}, \mathcal{R}$ values for $mcs = 10$ and $mcs = 20$. We emphasise that the

network quality remains unaffected and exhibits low std. In *Ab-2)* (middle rows), we observe that lowering the sphere coverage parameter $B$ significantly reduces the number of edges in $\mathcal{G}_{DD}$, while not affecting the resulting scores. This again empirically shows that the edge filtering, introduced in Section 3.2, correctly removes unstable edges even in higher dimensional spaces. Similarly, in *Ab-3)* (bottom rows), increasing the number of sampled rays $T$ naturally increased the number of found Delaunay edges which did not affect the resulting scores. These result demonstrate the reliability and stability of our DCA method even in higher dimensions, and show the efficiency of introduced edge filtering using $B$.

Table 8: DCA ablation results for StyleGAN obtained when varying $mcs$ in experiment *Ab-1)* (top rows), sphere coverage parameter $B$ in *Ab-2)* (middle rows) and number of sampled rays $T$ in *Ab-3)* (bottom rows) as described in Section B.2. For each experiment *Ab-i)*, we report in columns the obtained precision $\mathcal{P}$, recall $\mathcal{R}$, network quality $q(\mathcal{G})$, percentage of edges normalized by the maximum number obtained per experiment *Ab-i)* (with maximum absolute value shown in square brackets).

| | | $\mathcal{P}$ | $\mathcal{R}$ | $q(\mathcal{G})$ | % EDGES |
|---|---|---|---|---|---|
| | 3 | 0.992 | 0.970 | 0.494 | 1.0 [47093874] |
| | 5 | 0.954 | 0.900 | 0.495 | 0.896 |
| $mcs$ | 10 | 0.564 | 0.469 | 0.493 | 0.293 |
| | 20 | 0.564 | 0.469 | 0.493 | 0.293 |
| | | [0.768 ± 0.237] | [0.702 ± 0.271] | [0.494 ± 0.001] | |
| | 0.5 | 0.564 | 0.469 | 0.490 | 0.375 |
| | 0.6 | 0.564 | 0.469 | 0.492 | 0.537 |
| | 0.7 | 0.564 | 0.469 | 0.493 | 0.709 |
| $B$ | 0.8 | 0.564 | 0.469 | 0.493 | 0.859 |
| | 0.9 | 0.564 | 0.469 | 0.493 | 0.954 |
| | 1.0 | 0.564 | 0.469 | 0.4932 | 1.0 [13784316] |
| | | [0.564 ± 0.000] | [0.469 ± 0.000] | [0.492 ± 0.001] | |
| | 10 | 0.592 | 0.486 | 0.471 | 0.002 |
| | $10^2$ | 0.478 | 0.376 | 0.479 | 0.011 |
| | $10^3$ | 0.563 | 0.469 | 0.487 | 0.086 |
| $T$ | $10^4$ | 0.563 | 0.469 | 0.493 | 0.307 |
| | $10^5$ | 0.563 | 0.469 | 0.496 | 0.673 |
| | $10^6$ | 0.564 | 0.469 | 0.497 | 1.0 [44866410] |
| | | [0.553 ± 0.041] | [0.456 ± 0.040] | [0.487 ± 0.010] | |

## B.3    REPRESENTATIONS SPACE OF VGG16

In this section, we further describe the experiments performed to analyze the representation space of a VGG16 model (Simonyan & Zisserman, 2015) pretrained on the ImageNet dataset (Deng et al., 2009). We present the results obtained by repeating the experiment investigating separability of classes performed by Poklukar et al. (2021) and show additional examples of images analyzed in Section 4.3 where we applied q-DCA to investigate whether the limited class separation originates from inadequacies of the model or human labeling inconsistencies.

We used the same representations as Poklukar et al. (2021) who constructed $R$ and $E$ from representations of images corresponding to 5 ImageNet classes in each set. In version 1, classes were manually chosen to maximize the semantic differences in representations where $R$ and $E$ contained classes representing kitchen utilities and dogs, respectively, while in version 2 classes were chosen at random.

**Class separability** We analyzed the distilled Delaunay graph $\mathcal{G} = \mathcal{G}_{DD}(R \cup E)$ built on the sets $R$ and $E$ in each version. In Table 9, we report the DCA scores, the relative size of the largest connected component $|\mathcal{G}_0|_{\mathcal{V}}^r$ (normalized by $|R \cup E|$) and the total number of non-trivial components denoted as # *non-trivial*. In version 1, we observed one large connected component containing $\approx 61\%$ of all the points in $R$ and $E$, and 4 more non-trivial components containing $E$ points only which in turn yielded a lower precision $\mathcal{P}$. Consequentially, all the heterogeneous edges (among points from $R$ and $E$) contributing to the network quality $q(\mathcal{G})$ were contained in the first component. A relatively

comp. idx 1      comp. idx 2      comp. idx 3          comp. idx 1

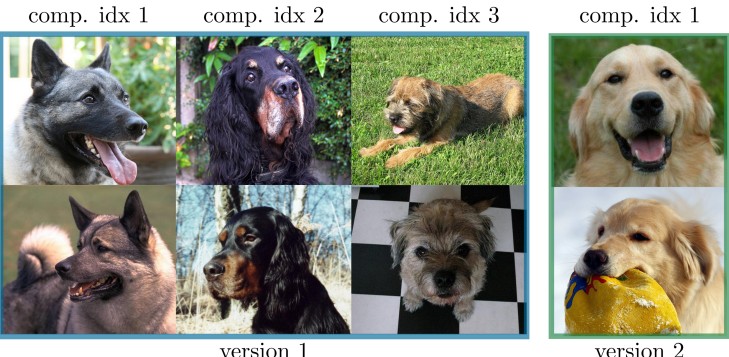

version 1               version 2

Figure 11: Examples of images contained in three non-trivial $E$ components obtained in version 1 (marked with blue) and in the single non-trivial $E$ component obtained in version 2 (marked with green).

large $\mathcal{R}$ suggests that features corresponding to images of kitchen utilities were encoded close by in the representation space. On the other hand, in version 2, we observed that the relative size $|\mathcal{G}_0|^r_{\mathcal{V}}$ and $q(\mathcal{G})$ increased compared to version 1 indicating that points in $R$ and $E$ are better aligned. Similarly as in version 1, we observed only one more non-trivial component again containing $E$ points only which thus did not contribute to $q(\mathcal{G})$ score. Moreover, we observe larger $\mathcal{P}$ and lower $\mathcal{R}$ compared to version 1 which means that more $E$ but less $R$ points were contained in the largest component due to the random choice. Interestingly, by visual inspection (see example images in Figure 11), we see that all non-trivial $E$ components in both versions consist of individual dog breeds suggesting that features representing dogs might be easier for the network to distinguish.

For comparison, we included GeomCA scores reported by Poklukar et al. (2021). We emphasise two main differences in the interpretation of the results: *i)* their network quality $q$ and consistency $c$ scores are affected by the sparsification process, while ours reflect the structure of the original sets $R$ and $E$, *ii)* their absolute values of $\mathcal{P}, \mathcal{R}$ and $|\mathcal{G}_0|^r_{\mathcal{V}}$ are substantially lower than ours which is the result of lower number of edges obtained in GeomCA. For example, the ratio of the number of edges over the number of nodes in version 1 resulted in 0.006 in GeomCA and 673.5 in DCA. This also indicates that $\varepsilon$-edges constructed based on maximum Euclidean distance are not as expressive in higher dimensions as Delaunay ones, which is also seen from higher number of smaller non-trivial connected components obtained in GeomCA.

Table 9: DCA scores (top row) obtained on VGG16 representations of ImageNet images from version 1 (kitchen utilities vs dogs) and version 2 (random) compared with GeomCA scores (bottom row). For each method, we report in columns network consistency $c(\mathcal{G})$, network quality $q(\mathcal{G})$, precision $\mathcal{P}$, recall $\mathcal{R}$, the relative size $|\mathcal{G}_0|^r_{\mathcal{V}}$ of the largest component normalized by the total number of points in $R \cup E$ and lastly, the total number of non-trivial connected components (*# non-trivial*).

|  | VERSION | $c(\mathcal{G})$ | $q(\mathcal{G})$ | $\mathcal{P}$ | $\mathcal{R}$ | $|\mathcal{G}_0|^r_{\mathcal{V}}$ | # NON-TRIVIAL |
|---|---|---|---|---|---|---|---|
| DCA | 1 | 0.98 | 0.31 | 0.4975 | 0.7247 | 0.610 | 5 |
|  | 2 | 1.0 | 0.42 | 0.6551 | 0.6302 | 0.640 | 2 |
| GEOMCA | 1 | 0.75 | 1.00 | 0.0042 | 0.0130 | 0.004 | 7 |
|  | 2 | 0.98 | 1.00 | 0.0423 | 0.0391 | 0.023 | 25 |

**Amending labelling inconsistencies** We show additional examples of images supporting the experiment performed in Section 4.3 where we applied q-DCA to further investigate the origin of the limited class separation, i.e., whether it emerged from inadequacies of the model or inconsistencies of human labeling. We first ran DCA on randomly sampled 5000 points from each of the $R$ and $E$ sets in both versions, and then determined the labels of the remaining 2758 and 3000 points in version 1 and 2, respectively, by the label of their closest point in the distilled Delaunay graph $\mathcal{G} = \mathcal{G}_{DD}(R \cup E)$ using q-DCA extension.

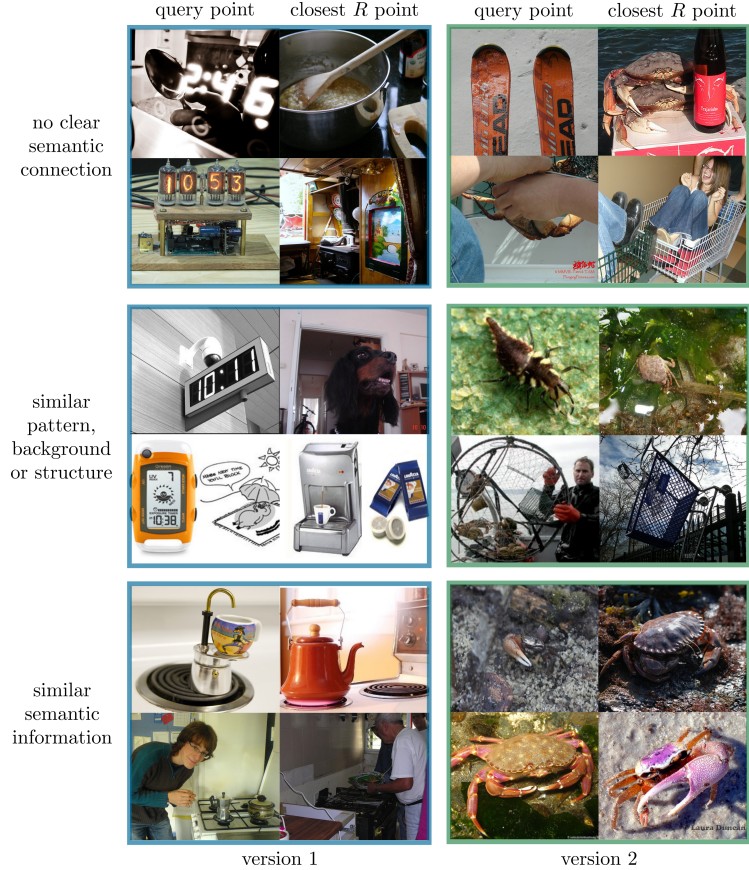

Figure 12: Additional examples of query images (odd columns) and their corresponding closest R images (even columns) taken from version 1 (left) and version 2 (right).

This approach resulted in $87.6\%$ accuracy in version 1, and $90.3\%$ in version 2, providing interesting insights into the nature of both human labeling and structure of the VGG16 representation space. In Figure 12, we show examples of query images (odd columns) and the corresponding closest image in $\mathcal{G}$ (even columns) *having different labels* corresponding to version 1 (left, blue) and version 2 (right, green). Similarly as in Figure 8, the first row shows examples of pairs of images considered close by VGG16 despite no clear semantic connection between them. The second row shows examples of images that are close in the VGG16 representation space possibly due to similar pattern, background or structure, while having meaningfully different labels. The last row shows examples where human labeling is different despite the images having strikingly similar semantic information, which is why they are encoded close by the VGG16 model.

