# OpenReview forum: "Delaunay Component Analysis for Evaluation of Data Representations"
_ICLR.cc/2022/Conference — ICLR 2022 Poster_

### Official Review · Reviewer_vso9 · 2021-10-29

**Correctness:** 4
**Technical Novelty And Significance:** 3
**Empirical Novelty And Significance:** 3
**Recommendation:** 8
**Confidence:** 4

**Main Review:**

Novelty and contribution:
- The proposed method has a clear motivation and an inbuilt superior robustness: knn or epsilon graphs are inconsistent for outliers and varying sampling densities, which DCA circumvents. The former also have an inbuilt hyperparameter (k, eps) which is difficult to choose in practice.
- The main contribution of the method is demonstrating that such graphs are a useful approximation of features manifolds. The main technical challenge is combining the established Delaunay graph (Polianskii & Pokorny, 2019) extraction method with the clustering algorithm of (McInnes et al., 2017).

Presentation:
- The overall presentation is adequate, but not always perfectly clear/easy to follow.
- Chapter 4 is overall too long and cluttered minor details. I suggest to move some of these details to the appendix and focus on highlight the core message of each experimental setting. This is especially crucial, since assessing the performance of the proposed method is generally not as simple as reading off a number. Advantages over prior methods need to be highlighted more clearly.
- The presentation would benefit from a brief, rigorous problem formulation paragraph at the beginning of Sec. 2.
- To make the paper more self-contained, I would like to see more details on the Delaunay graph extraction in Sec 3., since it is a key technique. Most details are currently in the appendix.

Method and results:
- The motivation of the method is clear and it is technically sound.
- I imagine that constructing the Delaunay graph for high-dimensional embeddings gets increasingly difficult. In practice, the number of Monte-Carlo sampled rays should increase with the number of dimensions and the graph becomes more connected (ergo the need for the pruning in Sec. 3.2). I would to see a more thorough discussion of this effect, the authors simply state that they usually choose T=1e4. How much does the sampling affect the results in higher dimensions? Error bars are only provided for the low-dim. embeddings in Sec. 4.1 and not for the high-dim. embeddings of StyleGAN and VGG16 (Sec. 4.2 and 4.3).
- The metrics P and R have a fairly coarse granularity and introduce additional hyperparameters eta_c, eta_q. What additional value do they provide, beyond c and q? Why not simply report the mean c and q values?
- Why do we need the distinction between DCA and q-DCA (in Sec. 3.1)? What is the difference of q-DCA and applying DCA with E={q}?

Minor comments:
- It is not clear from the paragraph "Phase 2" alone, whether HDBSCAN has additional tunable parameters, please add a short clarification.
- Fig. 1 needs a legend for the blue and orange nodes and edges, as well as a more detailed explanation of why some of the edges are solid and others dashed.
- Figure captions are generally not sufficient to interpret the figures throughout the paper.
- In Fig. 6, labels and colors are small and hard to distinguish. Do q^SG and c overlap closely? Moreover, P and R should be compared in separate graphs.
- Similar issues with Fig. 9 and Fig. 10 of the appendix.

=============================================================================================

Post discussion:
Reading the other reviews and the authors' responses did not raise any further, major questions on my end. The authors addressed most of my concerns in their response and evidently took the time to implement many of the reviewers' suggestions in the updated draft of the paper. I am therefore happy to increase my rating to "accept". I believe that the method has a strong motivation and the presented results are encouraging.

**Summary Of The Paper:**

The paper presents a novel approach to analyzing and comparing feature manifolds, e.g. from a learned embedding. The main idea is to construct a graph from a set of discrete samples to approximate the underlying, continuous manifold. The graph embedding can then be used to assess the similarity of a training and test set embedding, defined as their overlap and interconnectedness. Related existing methods construct such graphs via k-nearest neighbor or epsilon proximity graphs, whereas the proposed method uses Delaunay graphs.

**Summary Of The Review:**

The idea of using (pruned) Delaunay graphs to approximate continuous feature spaces is novel and technically sound. In my view, it has a lot of potential utility for analyzing high-dimensional embedding spaces, an important open problem. I am further convinced that this formulation is an improvement over related approaches such as IPR (Kynkaanniemi et al., 2019) or GeomCA  (Poklukar et al., 2021). On the other hand, several issues with the clarity of the presentation (especially in Sec. 4) obscure this main message. Overall, I am inclined to give the paper a more positive rating and hope that the authors can provide clarifications.

---

> ### Author Response · Authors · 2021-11-15
> **Presentation**
>
> We thank the reviewer for all the detailed suggestions on how to improve the clarity of the presentation. As stated in the general comment above, we will modify the submission to expand Section 3 with more details regarding the method as well as simplify Section 4 where we will highlight and clarify the main message of each experiment. We will additionally address as many minor concerns regarding the presentations as possible. We will provide a detailed summary of the changes once we submit the updated version of the paper.

---

> ### Author Response · Authors · 2021-11-15
> **GeomCA optional parameters**
>
> The $\eta_c$ and $\eta_q$ parameters inherited from the GeomCA scores [1] are considered as optional parameters that can be adjusted depending on the application. As pointed out by Poklukar et al [1], the role of these parameters is to provide the flexibility to consider only components that have certain consistency and quality properties, i.e., to adjust the definition of fundamental components. An example showcasing the benefits of these optional parameters is the contrastive learning setup in Section 4.1. where we expect the components to be of high quality and consistency if the contrastive learner was trained successfully. By setting higher thresholds for $\eta_c$ and $\eta_q$, we can efficiently perform the analysis and determine the quality of the contrastive learner. Note that these optional parameters affect only the resulting precision $\mathcal{P}$ and recall $\mathcal{R}$ scores but not the global network consistency $c(\mathcal{G})$ and network quality $q(\mathcal{G})$.
>
> [1] Petra Poklukar, Anastasiia Varava, Danica Kragic, GeomCA: Geometric Evaluation of Data Representations, International Conference on Machine Learning. PMLR, 2021.

---

> ### Author Response · Authors · 2021-11-15
> **Distinction between DCA and q-DCA**
>
> We thank the reviewer for pointing out the concern regarding the differences between DCA performed with $R$ and $E = \\{ q \\} $ and $q$-DCA. There are three differences between the two versions, two theoretical and one practical.
>
> When performing DCA on $R \cup \\{ q \\} $, we would build the Delaunay graph $\mathcal{G_{D}}(R \cup  \\{ q \\} )$ on $R \cup \\{ q \\} $ and then perform the distillation from Phase 2 which removes some of its edges and distills $\mathcal{G_{D}}(R \cup  \\{ q \\}  )$ into a graph $\mathcal{G_{DD}}(R \cup \\{ q \\})$ having several connected components. In contrast, one can think of performing $q$-DCA on a query point $q$ with respect to the graph $\mathcal{G_{DD}}(R)$ as having more steps: (i) we build the graph $\mathcal{G_{D}}(R)$ and distill it to obtain $\mathcal{G_{DD}}(R)$, (ii) we build $\\mathcal{G_{D}}(R \cup  \\{ q \\})$ and extract the neighbourhood $N(q)$ in that graph, and (iii) we intersect $N(q)$ with the distilled graph $\mathcal{G_{DD}}(R)$. Note that in practice, step (ii) is efficiently implemented by performing the Monte-Carlo sampling from Phase 1 only for the Voronoi cell $\text{Cell}(q)$.
>
> To illustrate the two theoretical differences, let us look at two examples. The first is when the query point lands close to two connected components in $\mathcal{G_{DD}}(R)$. Imagine, for example, that the outlying orange point in Figure 1 is our query point $q$. In this case, running DCA on $R \cup \\{ q \\}$ results in $q$ being marked as an outlier without any edges. However, if we were to insert this point using $q$-DCA, we would consider the neighbourhood $N(q)$ intersected with distilled graph $\mathcal{G_{DD}}(R)$ which would yield the long edges initially removed in DCA.
>
> For the second example, assume that $q$ lands within a connected component in $\mathcal{G_{DD}}(R)$. Imagine that $q$ is added on one of the blue interior edges in the bottom left connected component in Figure 1. The approximate Delaunay graph $\mathcal{G_{D}}(R \cup  \\{ q \\})$ built on $R \cup  \\{ q \\}$ would not contain the blue interior edge where $q$ lands and would instead contain edges to the neighbouring points of $q$. In other words, considering $q$ in building $\mathcal{G_{D}}(R \cup \\{ q \\} )$ can break some of the edges contained in the Delaunay graph built on $R$ only. These edges might be important for the subsequent analysis of $N(q)$ depending on the application.
>
> The last example also illustrates the practical difference between DCA and $q$-DCA, namely, that in $q$-DCA we consider $\mathcal{G_{DD}}(R)$ as the prior knowledge which we use to assess the quality of the query point $q$. Moreover, in cases where the query set $Q$ contains more than just one point, i.e., $|Q| > 1$, performing $q$-DCA bypasses building of the Delaunay graph $\mathcal{G_{D}}(R \cup  \\{ q \\} )$ for each query point $q$. Instead, $\mathcal{G_{D}}(R)$ and $\mathcal{G_{DD}}(R)$ are built once and we only compute the insertions of each $q$ in $\mathcal{G_{DD}}(R)$ by performing the Monte-Carlo sampling from Phase 1 only for the Voronoi cell $\text{Cell}(q)$.

---

> ### Author Response · Authors · 2021-11-15
> **Construction of the Delaunay graph in high dimensions**
>
> We agree with the reviewer that the current version of the paper is missing an ablation study on the $T$ hyperparameter in higher dimensions. To empirically support our answers below, we will modify the submission to include an ablation study on the tunable hyperparameters of DCA (similar to Table 5 in Appendix B.1) performed on high-dimensional representations of the StyleGAN model studied in Section 4.2. The results show that our method is stable with respect to variations in the hyperparameters even in higher dimensions, which was a concern shared across all the reviewers.
>
> Note that the number of edges naturally increases with an increased number of rays $T$ used in the Monte-Carlo sampling method proposed by Polianskii & Pokorny. However, this does not affect our scores which exhibit very low variance with respect to variations in $T$ (nor other hyperparameters). To better highlight this fact, we will add a row in Table 5 and in the newly added table including the ablation study in higher dimensions with the mean and standard deviation obtained when varying each hyperparameter. We emphasise, however, that the error bars in Figure 5 corresponding to the experiment in Section 4.1 are obtained with respect to variations in points for fixed values of DCA hyperparameters. On the contrary, the resulting scores in the ablation study are obtained with respect to variations in hyperparameters for a fixed set of points.

---

> ### Author Response · Authors · 2021-11-15
> **General answer**
>
> We thank the reviewer for the correct summary of our method as well as the detailed comments both regarding the method and presentation. For the sake of easier discussion, we answered each concern in a separate comment below.

---

> > ### Author Response · Authors · 2021-11-16
> > **New submission**
> >
> >
> > We would like to inform the reviewer about the updated submission of our paper with integrated feedback received from all the reviewers. Please see the general comment above for the exact details regarding the changes in the new version of the paper.

---

> > ### Comment · Reviewer_vso9 · 2021-11-19
> > **Post discussion comments**
> >
> > I thank the authors for their thorough response and for the updated version of the paper. I change my rating to "accept", see "Post discussion" for further comments

---

> > > ### Author Response · Authors · 2021-11-20
> > > **Thank you for the feedback**
> > >
> > > We thank the reviewer for the feedback. We updated the submission where we removed the orange color marking the changes.

---

### Official Review · Reviewer_vfbH · 2021-11-02

**Correctness:** 4
**Technical Novelty And Significance:** 2
**Empirical Novelty And Significance:** 3
**Recommendation:** 8
**Confidence:** 3

**Main Review:**

Pros.
- The paper is very well written and well motivated (especially the parts that show difficulties with GCA and prior methods)
- I find the evaluation to be comprehensive, especially the analysis in Section 4.


Cons (minor)
- (novelty) Conceptually, it appears that this paper builds on the main ideas proposed in GCA (which is not necessarily a bad thing). However, the main idea of opting for a delaunay graph is an act of making an existing algorithm (GCA) more robust without fundamentally proposing a new metric.

- The complexity of constructing the delaunay graph in high dimensions can be a computationally expensive affair.


**Summary Of The Paper:**

This paper explores a new method (DCA) to evaluate the quality of learned data representations. The goal of this paper is to score representations from sets R (reference) and E (evaluation) using geometric and topological properties (like connected components) of the representations spaces. To put simply, if the local geometry and overall topology of both R and E are similar, then the representations are considered consistent and aligned (hence yielding overall high precision and recall).

The authors primarily compare with 3 methods: GS, IPR and GCA (including a sparse variant) and argue that in scenarios of prevalent outliers and varying component densities of the representation spaces, such methods typically yield non-informative and suboptimal scores to compare representations. To that aid, the main method proposed in this paper proposes to alleviate these drawbacks and comprises of 3 steps: 1.) manifold estimation, 2.) component distillation 3.) component evaluation.

The main contribution of this paper is the use of Delaunay graphs in step (1.) as opposed to eps-neighbor or knn graphs as is done in prior methods. The authors claim that this choice of graph construction along with the distillation procedure leads to identifying consistent and high quality connected components leading to more stable results. In addition the authors also propose a ‘out-of-sample’ variant of their method to handle individual queries.

The authors demonstrate their metric in three different setups: (A.) Contrastive Learning on the synthetic Chamzas et al. (2021) images dataset (B.) Style GAN (C.) Representation space of VGG16. Results, especially on (A.) show that the proposed evaluation metric is better suited with accurate recognitions of mode-collapse and mode discovery.


**Summary Of The Review:**

All in all, I perceive this to be a good paper. The authors do a good job of highlighting drawbacks of prior methods to compare representation spaces and proposing a solution that mitigates them. The use of Delaunay graphs in lieu of k-nn based or eps-proximity based methods is an interesting way to explore the geometry and topology of representation manifolds. The diversity and comprehensiveness of evaluation is also quite good.

---

> ### Author Response · Authors · 2021-11-15
> **General answer**
>
> We thank the reviewer for the precise summary of our method as well as for recognizing the value of our improvements.
>
> We acknowledge the reviewer’s comment that our framework builds upon the three existing state-of-the-art methods, i.e., the Monte-Carlo method [1] that efficiently builds Delaunay graphs in higher dimensions, the HDBSCAN hierarchical clustering algorithm [2] for extracting the connected components of the obtained graph and GeomCA evaluation scores [3] for the final analysis of the connected components. We see the novelty of our contribution as two-fold: (1) conceptual, where we exploited the observation that Delaunay graphs better approximate data manifolds given various challenging geometric arrangements of the data points. Furthermore, we applied and adjusted the method of Polianskii & Pokorny [1] from the field of computational geometry to the machine learning setting, and (2) algorithmic, where we adjusted the above mentioned methods that each address very different problems to implement an efficient evaluation framework.
>
> Since the concern regarding the complexity of the framework was raised by all reviewers, we will modify our submission to include both (i) the theoretical analysis of the complexity of building Delaunay graphs, and (ii) empirical analysis on the performance of the DCA method in higher dimensions.
>
> Regarding (i), the computational complexity of the Delaunay graph construction grows polynomially with dimensionality $N$ in the proposed methodology. Given that the number of points $|R|+|E|$ is at least linear corresponding to their dimensionality $N$, i.e., $|R|+|E| = \Omega(N)$, we utilize the probabilistic method of graph construction proposed by Polianskii & Pokorny [1] that has complexity $\mathcal{O}((|R|+|E|)^2 \cdot (N+T))$. Moreover, we extend their method with a procedure that discards the least relevant edges as determined by the sphere coverage parameter $B$. In contrast, the classical exact Delaunay construction methods have complexity exponential to dimensionality and indeed are not applicable in practice for high-dimensional data.
>
> Regarding (ii), we will modify the submission to include the same ablation study as reported in Appendix B.1 but performed on high-dimensional representations of the StyleGAN model studied in Section 4.2. The results show that our method is stable with respect to variations in the hyperparameters even in higher dimensions. Please see the general comment above for more details on the planned changes to the submission.
>
>
>
> [1] Vladislav Polianskii, Florian T. Pokorny. Voronoi boundary classification: A high-dimensional geometric approach via weighted monte carlo integration. International Conference on Machine Learning. PMLR, 2019.
>
> [2] Leland McInnes, John Healy, Steve Astels. Hdbscan: Hierarchical density based clustering. Journal of Open Source Software 2.11 (2017): 205.
>
> [3] Petra Poklukar, Anastasiia Varava, Danica Kragic, GeomCA: Geometric Evaluation of Data Representations, International Conference on Machine Learning. PMLR, 2021.

---

> > ### Author Response · Authors · 2021-11-16
> > **New submission**
> >
> >
> > We would like to inform the reviewer about the updated submission of our paper with integrated feedback received from all the reviewers. Please see the general comment above for the exact details regarding the changes in the new version of the paper.

---

### Official Review · Reviewer_JKBs · 2021-11-12

**Correctness:** 3
**Technical Novelty And Significance:** 3
**Empirical Novelty And Significance:** 3
**Recommendation:** 6
**Confidence:** 3

**Main Review:**

Pros: (appearing in random order)

- Section 2 is well written and offers a clear picture of the limitations of existing methods (GS, IPR, GeomCA). In particular, the drawbacks deriving from either using $\epsilon$-proximity or $k$-NN approximations is well documented and rightly leads to finding better approximations to deal with outliers or varying density clusters.

- The structure of the DCA algorithm described in Section 3 is clear. The usage of Delaunay neighbourhood graphs seems a sound way of tackling the main limitations highlighted previously. The query point insertion and evaluations represents a consistent way of relying on the cell formalism to adapt to cases where data are being collected continuously. The proposed method to prune the graph based on the solid angles also seems geometrically clear.

- Efforts are made to test DCA on different scenarios in Section 4 to support the claims of better robustness and ability to adapt to cases with outliers.


Cons: (appearing in random order)

- While limitations of existing algorithms are thoroughly discussed, an intuitive picture of the potential of using Delaunay graphs is only briefly touched upon in Section 3, implicitly referring to the evaluation section to support the idea. Are there other ways of accounting for spatial information (neighbours)? What is particularly convenient in the Delaunay formalism?

- The novelty of the paper is not too strong. DCA amounts to linking existing methods together. While this in principle is not a bad thing, it seems that there isn't a significantly new idea somewhere in the paper about the different aspects of DCA being used, not even in the metric/component section.

- One of the major weaknesses is that the paper lacks analysis of complexity of the proposed method as opposed to the existing ones. This should be investigated to some extent. While the pruning method implicitly tries to address that, it is not clear the tradeoff between accuracy and complexity. In this regard, it is also not clear the role played by the number of sampled rays T, this should be somehow discussed.

- One of the main motivation for relying on DCA is avoiding difficult to tune hyperparameters. It seems though that in the distillation phase the clustering algorithm introduces a hyperparameter? Why is this more interpretable than, say, $\epsilon$ in the proximity graph construction? Similar questions for $\eta_{c}, \eta_{q}$.

- Section 4 is detailed but quite hard to access. The presentation here is not particularly good with results on the experiments that are very hard to understand with the exception - to some extent - of 4.1.

- (Minor) It would be good to understand whether the tradeoff in complexity is worth it to account for outliers. How much of an issue is that in general when we have high-dimensional data compared to the complexity of the method?

- (Minor) Could the pruning method proposed in 3.2 be potentially damaging for outliers? Also you propose to remove the longest edges such that the sum of solid angles corresponding to the remaining ones is \emph{larger} than a predetermined parameter B. Am I missing something here or should it be smaller than some parameter?





**Summary Of The Paper:**

This paper proposes a new method to assess the quality of learned data representations. Recent works have proposed to evaluate data representations by looking at geometric and topological alignment of a set of evaluation representations E and a set of reference set of representations R. The key idea in this paper consists in improving approximation of the data manifold using Delaunay neighbourhood graphs (DCA). The suggested method is meant to work better than existing algorithms (GS, IPR, GeomCA) in heterogeneous settings with outliers and/or varying cluster densities.

The DCA algorithm then amounts to realizing the Delaunay graph on the set $R\cup E$ relying on a Monte-Carlo method proposed in Polianskii & Pokorny (2019), distilling components using the hierarchical clustering algorithm HDBSCAN (McInnes
et al., 2017), and finally adopting the metrics introduced in Poklukar et al. (2021). Options to deal with query point extensions and pruning of the Delaunay graphs are discussed. Evaluation of the proposed DCA method are analysed for contrastive learning models trained with NT-Xent contrastive loss, for generation capabilities of a StyleGAN and on the VGG16 supervised model pretrained on the ImageNet dataset.



**Summary Of The Review:**

Despite my comments, I believe that the idea of relying on Delaunay neighbourhood graphs is worth investigating. The paper could improve on many aspects - presentation of the experiment section, intuition behind the proposed method, discussion on hyperparameters and most importantly complexity - however its core contributions are valid albeit with limited novelty.

---

> ### Author Response · Authors · 2021-11-15
> **Benefits of the Delaunay formalization**
>
> Delaunay graphs, and, more generally, Delaunay complexes, are widely used in computational topology as a standard tool for manifold approximation and shape reconstruction [1]. In contrast to the two main alternatives - KNN and epsilon-graphs, Delaunay graphs do not depend on hyperparameter tuning, while hyperparameters k in kNN and epsilon in epsilon-proximity graphs are crucial for defining neighbors. Moreover, they are often difficult to tune, especially when the underlying set of points is non-uniform and contains both dense and sparse regions.
>
> [1] Herbert Edelsbrunner, John Harer. Computational Topology: An Introduction. 2010
> Publisher: American Mathematical Society

---

> ### Author Response · Authors · 2021-11-15
> **Discussion on DCA hyperparameters**
>
> We thank the reviewer for pointing out the missing discussion on the hyperparameter T. We would like to direct the reviewer to our ablation study reported in Appendix B1, where we investigated the influence of all tunable hyperparameters (which are inherited from the aforementioned methods) in our framework, including T. Using the experimental setup from Section 4.1 with known ground truth, we showed that variations in $T$ do not affect our scores as they exhibit very low variance (see Table 5). In addition, we will modify the submission (see the general comment above) to include the same ablation study performed on high-dimensional representations of the StyleGAN model studied in Section 4.2. The results show that our method is stable with respect to variations in the hyperparameters even in higher dimensions, which was a concern shared across all reviewers.
>
> As mentioned by the reviewer, one of the tunable hyperparameters is the minimum cluster size $mcs$ associated to the HDBSCAN clustering algorithm. This parameter determines the minimum number of points that are required for forming a cluster, hence it directly influences the number and size of the connected components of the Delaunay graph. The approximate value (or range) of this parameter can often be inferred from the prior knowledge of the application, for instance, from the nature of the data. On contrary, the $\varepsilon$ hyperparameter in a proximity graph defines the maximum distance between two points connected by an edge. This is highly dependent on the ambient space. For example, the optimal value of the $\varepsilon$ parameter differs for embeddings on the same data in different neural networks layers. Moreover, obtaining a reliable estimate of $\varepsilon$ requires a priori computation of the distances among embeddings, which is indeed the approach adopted by Poklukar et al [1].
>
> The $\eta_c$ and $\eta_q$ parameters inherited from the GeomCA scores [1] are considered as optional parameters that can be adjusted depending on the application, similar to mcs. As pointed out by Poklukar et al [1], the role of these parameters is to provide the flexibility to consider only components that have certain consistency and quality properties, i.e., to adjust the definition of fundamental components. An example showcasing the benefits of these optional parameters is the contrastive learning setup in Section 4.1. where we expect the components to be of high quality and consistency if the contrastive learner was trained successfully. By setting higher thresholds for $\eta_c$ and $\eta_q$, we can efficiently perform the analysis and determine the quality of the contrastive learner.
>
> [1] Petra Poklukar, Anastasiia Varava, Danica Kragic, GeomCA: Geometric Evaluation of Data Representations, International Conference on Machine Learning. PMLR, 2021.

---

> ### Author Response · Authors · 2021-11-15
> **Discussion on outliers**
>
> We believe that accounting for outliers depends on the given use case of the framework, while computational complexity depends on the number of representations we wish to analyze. If outliers are not important to detect and analyze, one can resort to computationally simpler methods. However, in datasets with a large number of representations, prior methods such as IPR, GS and GeomCA are not necessarily computationally less expensive. Furthermore, representations of real world datasets often additionally form clusters of different shape and density which are better captured using Delaunay graphs compared to the approaches of the prior work. Even when the Delaunay graph is pruned using a $B < 1.0$, outliers remain unaffected. The pruning procedure keeps removing the longest edges originating from a fixed point until the sum of the spherical angles is larger than the chosen $B$ (as stated in the paper). In this way, we indeed remove the furthest neighbours first. However, some stable connections from outliers to other points will remain. Firstly, from the perspective of an outlier point, this procedure will not remove many of its edges. Secondly, we can expect that there will likely exist stable connections in high dimensions from regular points that will have a large enough spherical angle to not be removed. In other words, for these points, removing even the largest edge often results in the sum of spherical angles corresponding to the remaining edges that are below the $B$ threshold. Therefore, outliers remain in the obtained Delunay graph but later are marked as outliers by HDBSCAN.

---

> ### Author Response · Authors · 2021-11-15
> **Complexity analysis**
>
> We thank the reviewer for pointing out the missing analysis on the complexity of our approach. The general methodology, described in the paper, relies on adaptation of several pre-existing methods, namely [1] and [2], into our framework. Our changes to the algorithms do not alter the core complexity of those methods. The complexity of [1] is thoroughly discussed in the original paper. The full complexity is equal to $\mathcal{O}(|R| |E| N + |R| |E| T + (|R| + |E|) T N)$, where $N$ is the dimensionality of the representations and $T$ number of sampled rays. This may be further simplified as $\mathcal{O}((|R|+|E|)^2 (N+T))$ when the number of points is at least as large as their dimensionality; the complexity for q-DCA is obtained by substituting $|E|$ with $|Q|$. HDBSCAN’s asymptotics are more difficult to judge, and rely on the graph obtained in the step before. We can say that it does not directly depend on the dimensionality $N$ and is at most quadratic of the number of points, i.e. $\mathcal{O}((|R|+|E|)^2)$, as presented in a detailed discussion in [3-4].  We will add a brief analysis of complexity in the main paper to clarify how it corresponds to the changes in the algorithms.
>
> [1] Vladislav Polianskii, Florian T. Pokorny. Voronoi boundary classification: A high-dimensional geometric approach via weighted monte carlo integration. International Conference on Machine Learning. PMLR, 2019.
>
> [2] Leland McInnes, John Healy, Steve Astels. Hdbscan: Hierarchical density based clustering. Journal of Open Source Software 2.11 (2017): 205.
>
> [3] Leland McInnes, and John Healy. Accelerated hierarchical density based clustering. 2017 IEEE International Conference on Data Mining Workshops (ICDMW).
>
> [4] Ricardo JGB Campello, et al. Hierarchical density estimates for data clustering, visualization, and outlier detection. ACM Transactions on Knowledge Discovery from Data (TKDD) 10.1 (2015): 1-51.

---

> ### Author Response · Authors · 2021-11-15
> **General answer**
>
> We thank the reviewer for the correct summary of our method as well as all the detailed comments. For the sake of easier discussion, we answered each concern in a separate comment below. In the following days, we will modify the submission to address the concerns of all the reviewers as stated in the general comment above.
>
> We acknowledge the reviewer’s comment that our framework builds upon the three existing state-of-the-art methods, i.e., the Monte-Carlo method [1] that efficiently builds Delaunay graphs in higher dimensions, the HDBSCAN hierarchical clustering algorithm [2] for extracting the connected components of the obtained graph and GeomCA evaluation scores [3] for the final analysis of the connected components. We see the novelty of our contribution as two-fold: (1) conceptual, where we exploited the observation that Delaunay graphs better approximate data manifolds given various challenging geometric arrangements of the data points. Furthermore, we applied and adjusted the method of Polianskii & Pokorny [1] from the field of computational geometry to the machine learning setting, and (2) algorithmic, where we adjusted the above mentioned methods that each address very different problems to implement an efficient evaluation framework. Please note the aim of this work was not to improve the evaluation scores presented by Poklukar et al [3] which we believe are already adequately informative.
>
>
> [1] Vladislav Polianskii, Florian T. Pokorny. Voronoi boundary classification: A high-dimensional geometric approach via weighted monte carlo integration. International Conference on Machine Learning. PMLR, 2019.
>
> [2] Leland McInnes, John Healy, Steve Astels. Hdbscan: Hierarchical density based clustering. Journal of Open Source Software 2.11 (2017): 205.
>
> [3] Petra Poklukar, Anastasiia Varava, Danica Kragic, GeomCA: Geometric Evaluation of Data Representations, International Conference on Machine Learning. PMLR, 2021.

---

> > ### Author Response · Authors · 2021-11-16
> > **New submission**
> >
> >
> > We would like to inform the reviewer about the updated submission of our paper with integrated feedback received from all the reviewers. Please see the general comment above for the exact details regarding the changes in the new version of the paper.

---

> ### Comment · Reviewer_JKBs · 2021-11-19
> **Thanks for the revised version**
>
> I think the submission now is stronger than its original state and I believe there are no major concerns.
>
> However, increasing the score (necessarily to 8) would currently place the paper in the top 1% of all ICLR22 submissions. Although this is a good paper, $worthy$ of being accepted, I think that some truly novel contribution should be present in papers belonging to the top 1% - beyond adapting or combining known models/SOTA methods in a new direction. Accordingly, I have decided to stick to 6.

---

> > ### Author Response · Authors · 2021-11-20
> > **Thank you for the feedback**
> >
> > We thank the reviewer for the feedback. We updated the submission where we removed the orange color marking the changes.

---

### Author Response · Authors · 2021-11-15
**General answer to all reviewers**

We thank the reviewers for their thorough comments. We addressed in detail each of the reviewer’s concerns below. In this comment, we summarize the main concerns shared among all reviewers: (i) missing analysis of the performance of the DCA method in higher dimensions including the complexity analysis of the core part on building Delaunay graphs, and (ii) inadequate presentation of the performed experiments in Section 4.

In order to address these two main concerns, we will modify the submission in the following way:
- Expand Section 3 with details on Delaunay building algorithm including its complexity analysis,
- Expand Section 3 with explicitly stating the tunable hyperparameters of the algorithm, namely, T, mcs, and optional parameters B, eta_c, eta_q
- Simplify Section 4 by moving the cluster assignment experiment to the Appendix,
- Expand Section 4 with a more intuitive explanation of the results.

We believe that these changes will improve the presentation of our method as well as provide further evidence of its benefits. We will inform the reviewers when the submission is posted and provide the exact details regarding the changes in the comments to this post.

---

> ### Author Response · Authors · 2021-11-16
> **New submission**
>
> Dear reviewers,
>
> We updated the submission with a revised version of our paper where we addressed as many of the raised concerns as possible. We marked all our changes to the text with an orange color. In detail, we made the following changes:
> - Expanded Section 3 providing more details for Phase 1 and Phase 2 to make it more self-contained;
> - Added an extra subsection (Section 3.3) discussing the complexity of the core part of proposed framework;
> - Moved the cluster assignment experiment from Section 4.1. to the Appendix B.1;
> - Expanded the discussions in Section 4 with a more intuitive explanation highlighting the advantages of our approach and with extra empirical results supporting the conclusions;
> - Expanded Section 4.2 with a paragraph discussing the ablation study performed for high dimensional embeddings;
> - Expanded the captions of Figure 1 and added the legend;
> - Updated Figure 6 and Figure 10 and separately plotted precision and recall scores.
>
> Once again, we hope that these changes improved the clarity of our presentation and provided the clarification to the reviewers’ comments.

---

### Decision · Program_Chairs · 2022-01-20

**Decision:**

Accept (Poster)

**Comment:**

The paper proposes "Delaunay Component Analysis", a novel manifold learning technique. Reviewers raised several concerns regarding novelty, computational complexity of the method, and presentation. The authors provided a thorough rebuttal and engaged in discussion with the reviewers that addressed the concerns in a satisfactory manner. After the discussion, all the reviewers and AC recommend acceptance.